# Two-way communication between SecY and SecA suggests a Brownian ratchet mechanism for protein translocation

William John Allen[1†], Robin Adam Corey[1†], Peter Oatley[2,3†], Richard Barry Sessions[1], Steve A Baldwin[2,3‡], Sheena E Radford[2,4], Roman Tuma[2,4*], Ian Collinson[1*]

[1]School of Biochemistry, University of Bristol, Bristol, United Kingdom; [2]Astbury Centre for Structural Molecular Biology, University of Leeds, Leeds, United Kingdom; [3]School of Biomedical Sciences, University of Leeds, Leeds, United Kingdom; [4]School of Molecular and Cellular Biology, University of Leeds, Leeds, United Kingdom

**Abstract** The essential process of protein secretion is achieved by the ubiquitous Sec machinery. In prokaryotes, the drive for translocation comes from ATP hydrolysis by the cytosolic motor-protein SecA, in concert with the proton motive force (PMF). However, the mechanism through which ATP hydrolysis by SecA is coupled to directional movement through SecYEG is unclear. Here, we combine all-atom molecular dynamics (MD) simulations with single molecule FRET and biochemical assays. We show that ATP binding by SecA causes opening of the SecY-channel at long range, while substrates at the SecY-channel entrance feed back to regulate nucleotide exchange by SecA. This two-way communication suggests a new, unifying 'Brownian ratchet' mechanism, whereby ATP binding and hydrolysis bias the direction of polypeptide diffusion. The model represents a solution to the problem of transporting inherently variable substrates such as polypeptides, and may underlie mechanisms of other motors that translocate proteins and nucleic acids.

*For correspondence: r.tuma@ leeds.ac.uk (RT); ian.collinson@ bristol.ac.uk (IC)

†These authors contributed equally to this work

‡Deceased

**Competing interests:** The authors declare that no competing interests exist.

## Introduction

The Sec system is the main pathway for protein secretion in all forms of life. At the translocon core is a hetero-trimeric membrane protein complex – SecYEG in the plasma membrane of prokaryotes and Sec61αγβ in the ER of eukaryotes – which forms a protein channel across the membrane (*Ito et al., 1983*; *Brundage et al., 1990*; *Görlich et al., 1992*). Pre-protein substrates with an N-terminal signal sequence are targeted to the translocon in an unfolded state (*Arkowitz et al., 1993*), whereupon they are threaded across (secretion) or transferred laterally into the membrane (insertion). Protein translocation can occur either co-translationally following emergence of the nascent chain from the ribosome, or post-translationally. In prokaryotes, secretion is mostly post-translational (*Hartl et al., 1990*): substrates are recognised by the cytosolic ATPase SecA (*Lill et al., 1989*), which targets them to SecYEG and then drives them through the protein channel using energy derived from ATP binding and hydrolysis, and the trans-membrane proton motive force (PMF) (*Brundage et al., 1990*).

A structure of SecYEβ from *Methanococcus jannaschii* provided the first glimpse of the trans-membrane channel through which secretory proteins pass (*Van den Berg et al., 2004*) (PDB code 1RHZ; *Figure 1A*). SecY consists of ten trans-membrane helices (TMs), which form a 'clamshell' structure in the membrane, with five TMs on either side of the central protein channel, blocked by a 'plug' and a seal comprising six hydrophobic side chains (*Figure 1A*). This conformation is reinforced

**eLife digest** A protective membrane surrounds all cells, and controls what goes in and out of the cell. Many proteins that are made inside the cell need to be exported in order to do their job. In most organisms, a specialised transport motor that sits inside the membrane, known as 'Sec', carries out this export process. Sec recognises proteins that need to be exported and pushes them across the membrane and out of the cell. The energy required to do this comes from the cell's universal power source, a molecule called ATP.

Previous studies have shown what Sec looks like, but not how it pushes proteins from one side of the membrane to the other. Currently, the most popular theory for how Sec works is that it grabs hold of part of the protein and pushes it through a gate in the membrane. It then lets go and goes back to grab and push the next bit of the protein.

Allen, Corey, Oatley et al. have now used a combination of experimental and computational methods to look at how the different parts of Sec move around as it uses ATP. The reasoning behind using these methods was that it's easier to understand how a motor works by watching it in action rather than just looking at a still picture.

Using these methods, Allen, Corey, Oatley et al. show that the biggest movement in Sec as it uses ATP is in the membrane gate itself, which opens and closes. This suggests that Sec acts like a turnstile: proteins can freely move one way across the membrane, but are prevented from moving back in again. This mechanism has not been described before and may apply to other transport systems.

Further investigations will be needed to understand exactly how Sec recognises cargo and starts the transport process, and to explore the specific features of a protein that activate the turnstile. It also remains to be discovered how this transport process differs in other, non-bacterial cells. This could potentially help us develop new drugs that specifically block the bacterial Sec system without affecting human cells.

on one side by SecE, while on the opposite side a lateral gate (LG) connects the central protein-conducting channel to the lipid bilayer (*Figure 1A*). Thus, opening of the 'clamshell' could facilitate passage of secretory proteins through the central channel across the membrane, as well as the partitioning of membrane proteins *via* the LG into the bilayer.

SecA is a cytoplasmic ATPase that drives protein translocation through SecYEG (*Lill et al., 1989*; *Brundage et al., 1990*; *Akimaru et al., 1991*). The ATP turnover cycle of SecA is directly coupled to protein translocation (*Economou and Wickner, 1994*; *Karamanou et al., 1999*), but as yet the mechanism by which this is accomplished is unclear. Structural studies have revealed that SecA consists of two RecA-like nucleotide binding domains (NBD1 and NBD2), which together form the nucleotide-binding site (NBS) (*Hunt et al., 2002*) (*Figure 1B*). A 'two-helix finger' (2HF), at the SecY binding interface (*Zimmer et al., 2008*), and a pre-protein cross-linking domain (PPXD) both make contacts with the translocating polypeptide (*Bauer and Rapoport, 2009*).

A breakthrough in our understanding of how SecYEG and SecA operate came with a crystal structure of a complex of the two from *T. maritima* (*Zimmer et al., 2008*). SecY and SecA interact tightly, with the 2HF from SecA protruding into the SecY channel, and a long loop between TMs 6–7 of SecY buried inside SecA (*Figure 1B*). The structure also revealed conformational changes within both SecYEG and SecA. In SecA, a large movement of the PPXD activates the ATPase (*Gold et al., 2013*) and is thought to clamp the substrate in place, while in SecYEG the LG opens, causing a widening of the channel through which translocating substrates pass (*Figure 1B*) (*Zimmer et al., 2008*). Presumably, these changes prime the complex for intercalation of the pre-protein: however, they represent only a single snapshot in the dynamic process of protein translocation, regulated ultimately by the binding and hydrolysis of ATP. More recently, several electron microscopy structures have been solved for the Sec machinery in complex with the ribosome and nascent chains (*Park et al., 2014*; *Gogala et al., 2014*; *Pfeffer et al., 2015*; *Voorhees and Hegde, 2016*). However, whilst these structures reveal key features of the interactions between components, they do not provide mechanistic insights of how protein translocation *per se* occurs. Here, we take a

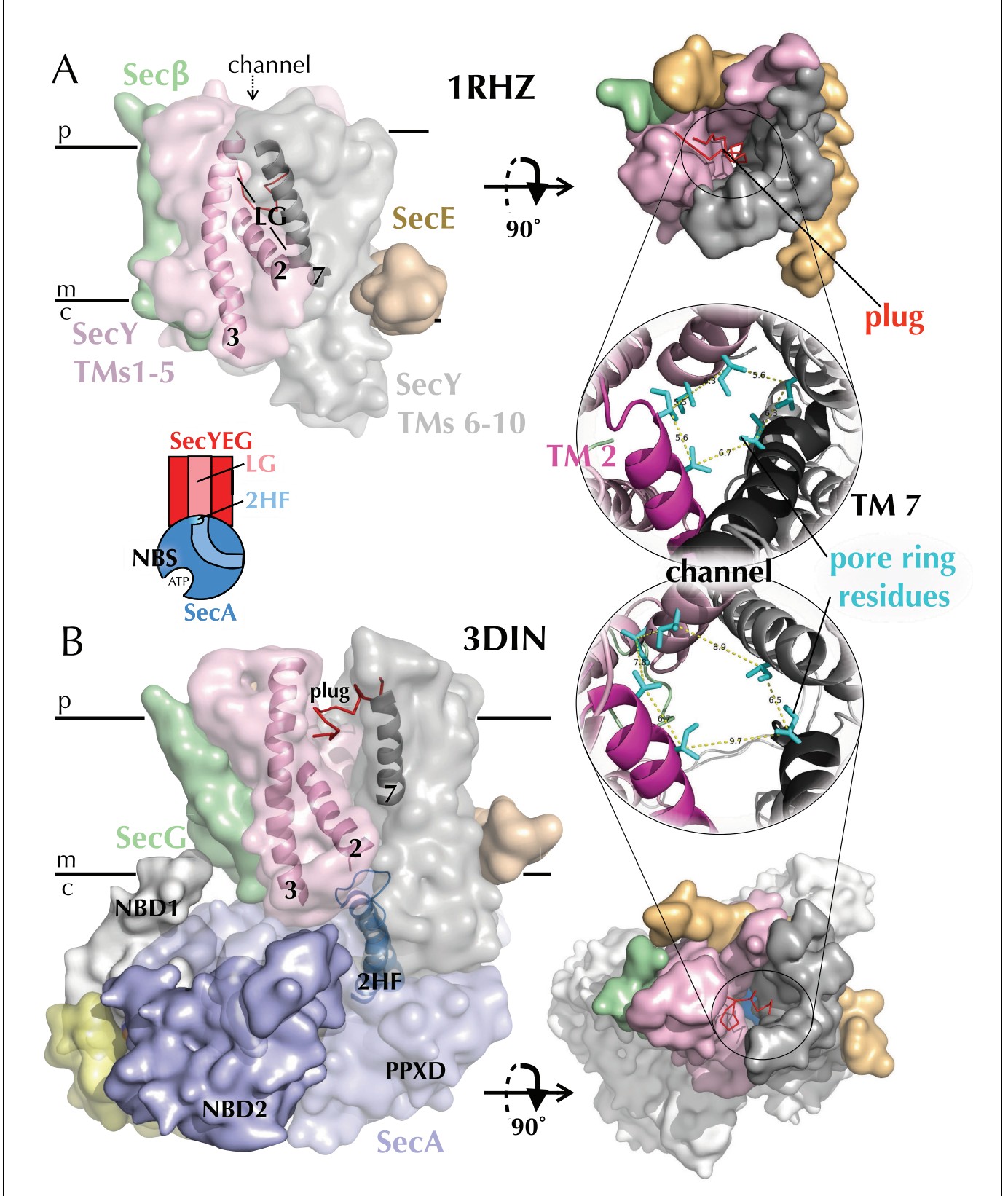

**Figure 1.** Structures of SecYEG. (A) Surface representations of SecYEβ from *M. janaschii*; PDB code 1RHZ (*Van den Berg et al., 2004*) viewed from the side (left) and periplasm (top right). TMs 1–5 of SecY are shown in pink and TMs 6–10 in grey, with the LG helices (TMs 2, 3 and 7) highlighted as
*Figure 1 continued on next page*

different approach: using all-atom molecular dynamics (MD) simulations, we identify key structural differences between the ATP- and ADP-bound complexes, and combine these with biochemical and single molecule FRET analyses. Together, the results allow us to propose a model for ATP-dependent transport, in which two-way communication between the NBS of SecA and the channel of SecY biases the direction of random substrate diffusion, in a 'Brownian ratchet' mechanism.

## Results

### Molecular dynamics reveals nucleotide-dependent structural changes in SecY

The structure of SecYEG-SecA was previously determined bound to a non-hydrolysable, non-natural nucleotide, ADP-BeF$_x$ (*Figure 1B*; PDB code 3DIN) (*Zimmer et al., 2008*). ADP-BeF$_x$ has been reported to mimic ATP in a pre-hydrolysis state (*Fisher et al., 1995*; *Phan et al., 1997*), however it has also been shown to produce post-hydrolysis intermediate states (*Peyser et al., 2001*). In addition, the presence of the fluoroberyllate ion has been known to produce artefacts (*Henry et al., 1993*). We reasoned that changing the bound nucleotide to either an ATP or ADP molecule *in silico* would help illuminate the two key states of the SecA hydrolytic cycle. To achieve this, complexes were built using the protein and Mg$^{2+}$ atoms from 3DIN, but with either ADP or ATP in place of ADP-BeF$_x$ (*Piggot et al., 2012*). The resultant structures were embedded in a POPC membrane (*Ulmschneider and Ulmschneider, 2009*) and five MD simulations of between 0.4–1 μs were run for each state (*Figure 2* and *Figure 2—figure supplement 1*).

When ADP-BeF$_x$ was replaced with ATP, the LG either retained its part-open ('PO') state of about 1.8 nm (*Figure 2A*, middle panel, compare to *Figure 2B*, lower panel; *Figure 2C*) or widened further to ~2.6 nm forming an open state ('O'; *Figure 2A*, lower panel; *Figure 2C*). Conversely, substitution of ADP-BeF$_x$ with ADP resulted in LG closure in all five simulations to ~1.5 nm ('C'; *Figure 2A*, upper panel; *Figure 2C*). Remarkably, this closed state closely resembles a previous structure of the resting SecY-complex in the absence of SecA (1RHZ (*Van den Berg et al., 2004*); *Figure 2B*, upper panel and *Figure 2C*). The distance over which this conformational change is propagated – over 5 nm from the NBS to the LG – is considerable. Note that simulations using a different molecular mechanics force field (Amber ff99SB-ILDN; *Figure 2—figure supplement 2*) show a similar effect.

The canonical helicase motifs within the NBS of SecA – particularly the conserved arginine finger in helicase motif VI, which coordinates the β-phosphate of the bound nucleotide (*Pause et al., 1993*) – respond differently to ATP and ADP (*Figure 3*). In this system the configuration of these residues correlates to the open or closed state of the channel (*Figure 3C*). Along with additional conformational changes (detailed in the legend to *Figure 3*), these data suggest a clear route of signal transduction from the SecA NBS to the polypeptide binding sites. However, the rest of the pathway within SecA is too subtle to be apparent from the MD data. Nonetheless, the initial opening or closing event is rapid: all the ADP and ATP trajectories diverge within the first 10 ns (*Figure 2C*). The speed of this suggests that the input model represents a high-energy intermediate – probably due to the presence of ADP-BeF$_x$.

### A FRET system monitors lateral gate opening

The stability of the three conformations observed (O, PO and C, *Figure 2—figure supplement 1*) suggests that they represent genuine metastable states, however MD sampling is not sufficient to determine their relative populations. To test the MD results experimentally, donor (Alexa Fluor 488) and acceptor (Alexa Fluor 594) fluorescent probes were attached to a unique cysteine pair flanking the LG (A103 and V353 of *Escherichia coli* SecY; *Figure 4A*; Cβ-Cβ distances 3.4–4.4 nm; the Förster radius, R$_0$, is 6 nm for the chosen dye pair). These allow the open and closed states of the LG to be distinguished by FRET. The doubly labelled protein (hereafter SecY**EG) is active (*Figure 4—figure supplement 1A*) and limited proteolysis confirms equal labelling on both sites (*Figure 4—figure supplement 1B+C*). After reconstitution into proteo-liposomes (PLs), about 50% (as judged by trypsin protection) of the complexes face outwards (*Figure 4—figure supplement 1D*), as has been observed previously for wild-type SecYEG (*Mao et al., 2013*; *Schulze et al., 2014*).

Equilibrium fluorescence measurements of SecY**EG show a decrease in FRET upon binding of SecA (*Figure 4B*), indicating LG movement – as expected from the crystal structures. A further FRET

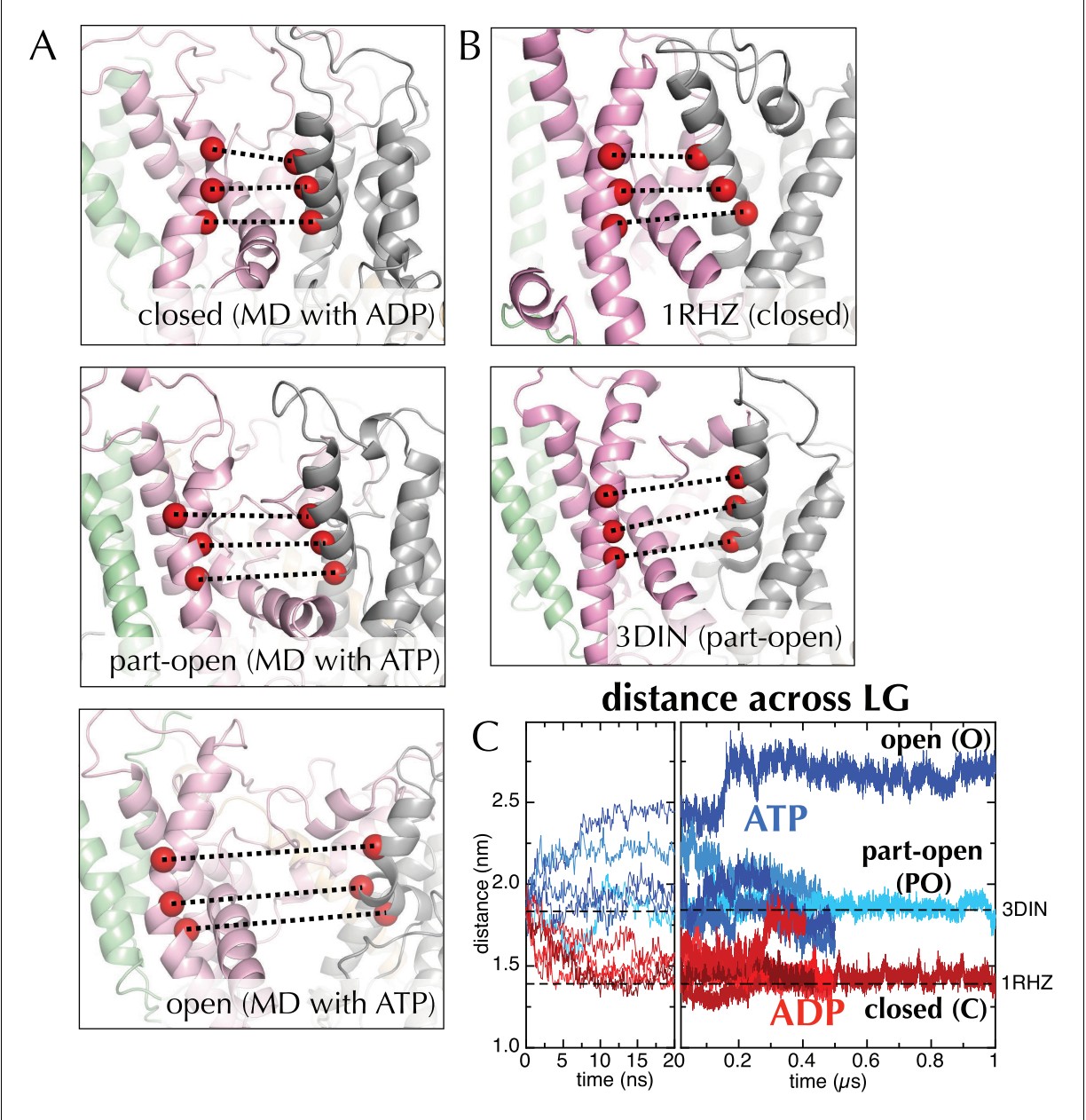

**Figure 2.** Molecular dynamics reveals nucleotide-dependent changes in the lateral gate of SecY. (**A**) 1 μs snapshots of the closed (top; after MD with ADP), part-open (middle, ATP) and open (bottom, ATP) forms of SecYEG, viewed side-on into the LG. TMs 1–5 of SecY are shown in pink, TMs 6–10 in grey, and SecG in green. Three residue pairs across the LG are shown (red spheres; residues 124/275, 127/278 and 130/282 in *T. maritima* numbering), with the distances between them drawn out as dotted black lines. Note that in the OPLS-AA force field, ATP and AMPPNP are essentially indistinguishable. (**B**) As in panel A, but showing the closed LG of SecYEβ (*M. janaschii*; PDB code 1RHZ (*van den Berg et al., 2004*); top) and the part-open LG of SecYEG-SecA (*T. maritima*, PDB code 3DIN (*Zimmer et al., 2008*); bottom). (**C**) Average distances between the residue pairs in panel B across all 10 MD simulations, with the initial 20 ns expanded for clarity. The input (3DIN) and the resting translocon (1RHZ) distances are also indicated. ATP simulations are shown in blue and ADP simulations in red, with representative open (O), part open (PO) and closed (C) trajectories indicated.

The following figure supplements are available for figure 2:

**Figure supplement 1.** Molecular dynamics stability.

**Figure supplement 2.** MD with Amber force field.

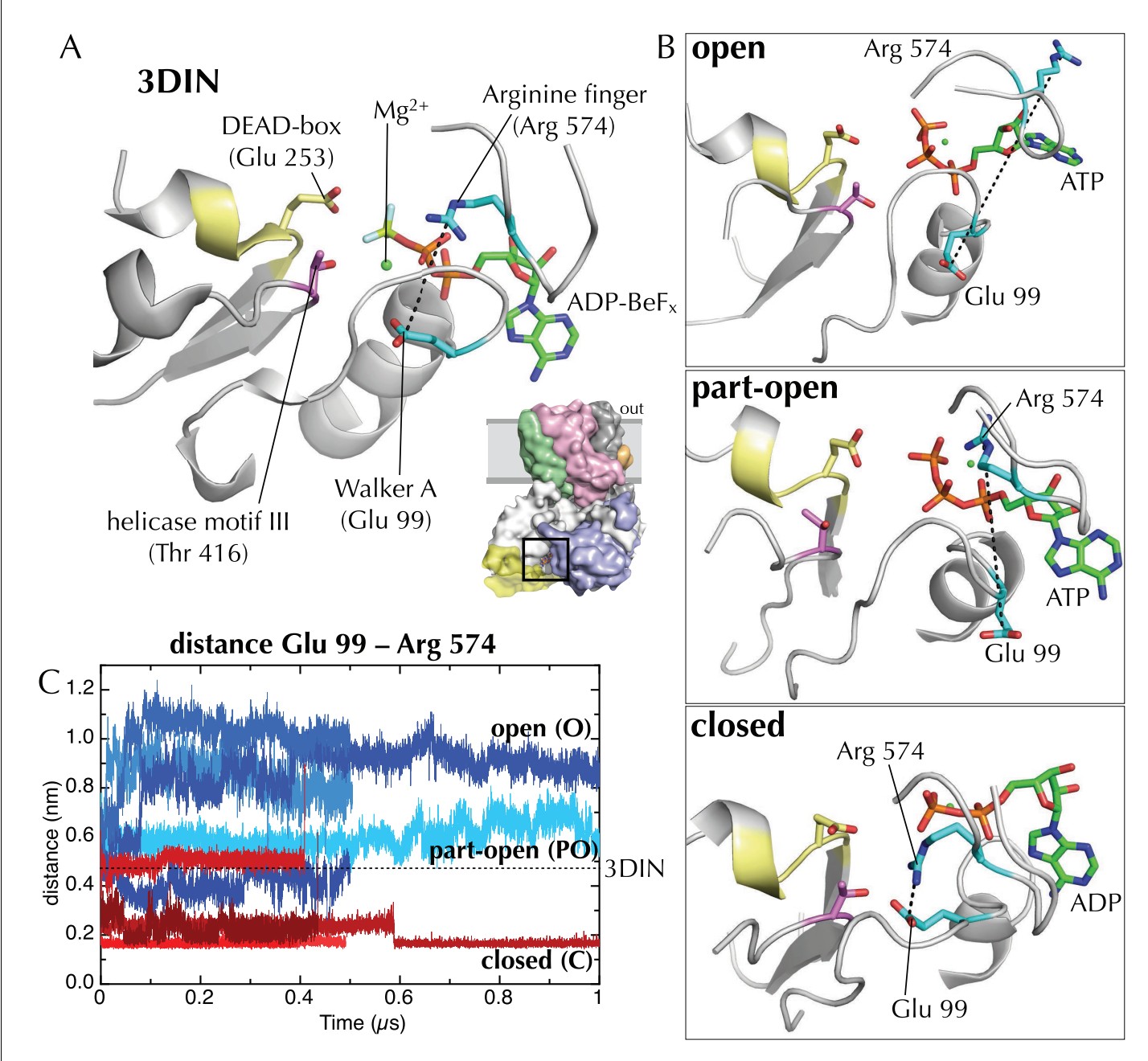

**Figure 3.** Effects of ATP and ADP on the nucleotide binding site. (A) Closeup of the NBS of SecA in the *T. maritima* SecYEG-SecA structure (3DIN; [*Zimmer et al., 2008*]).The ADP-BeF$_x$ moiety is shown as sticks (carbon green, nitrogen blue, oxygen red, phosphorus orange, beryllium lime and fluorine pale blue), and key helicase loops are shown as white cartoons. Conserved helicase features are shown as coloured sticks and numbered as in *T. maritima*: Glu99 of Walker A (helicase motif Ia); Glu253 of Walker B (DEAD box of helicase motif II); Thr416 of helicase motif III; and Arg574 ('arginine finger') of helicase motif VI. A black dotted line marks the distance between Arg574 and Glu99. (B) The same view as in panel A, but for post-microsecond simulation snapshots in the open, part-open and closed simulations. Of note are Arg574 and Glu99, which remain distant in the open and part-open simulations (ATP), but come close enough to form a salt bridge in the closed simulation (ADP). In helicases, this conserved arginine finger on motif VI has been shown to couple ATPase activity with substrate binding (*Hall et al., 1998*), underlining its importance in SecA. After the Arg574 to Glu99 salt bridge has formed in the ADP simulations, the Walker A motif is bought into contact with a conserved threonine (Thr416) in helicase motif III, which in the helicase family has been shown to use ATP binding to create an RNA binding site (*Banroques et al., 2010*). This threonine communicates directly with the SecA DEAD box—providing a likely route forsignal transmission of ATP hydrolysis away from the NBS towards the polypeptide binding sites. (C) Minimum distance analysis between residues Glu99 and Arg574 of SecA across the simulation data confirms that a Glu99-Arg574 interaction is formed in 4/5 of the ADP simulations, whereas it is absent in all 5 ATP simulations. Interestingly, the ADP simulation that does not find this salt bridge is

*Figure 3 continued on next page*

*Figure 3 continued*

the same simulation in which the SecY LG samples the part-open state (*Figure 2C*). Thus, it appears that nucleotide dependent formation and breaking of the Glu99-Arg574 salt bridge is involved in the coupling process leading ultimately to the respective closure and opening of the channel.

decrease is observed upon addition of the non-hydrolysable ATP analogue AMPPNP, but not ADP (*Figure 4B*). Titration of SecA into SecY**EG PLs in the presence of either ADP or AMPPNP (*Figure 4—figure supplement 1E*) shows that the changes in apparent FRET efficiency are not caused by differences in affinity between SecYEG and SecA in different nucleotide states, but rather represent differences in the width of the LG, *i.e.* a closure of the LG in the presence of ADP on SecA, some 5 nm away – consistent with the MD results.

## Single molecule analysis of the opening and closure of the lateral gate

Ensemble FRET measurements cannot be interpreted easily in terms of mechanistic details, because they report on the average of a mixed population of molecules. Only a fraction of SecYEG molecules are expected to be oriented correctly to engage substrate and labelled such that they give rise to a FRET signal (*Figure 4—figure supplement 2*), thus specific FRET changes can be drowned out in the ensemble signal.

To overcome this, we used single molecule FRET (smFRET). This technique allows the conformation of the LG of SecYEG to be monitored in the presence of SecA and different nucleotides, and can reveal the existence and proportions of different conformations within the ensemble. PLs containing single copies of SecY**EG were tethered to a glass supported bilayer as previously described (*Figure 5A*) (*Deville et al., 2011*). From time-lapse TIRF images, particles containing a single FRET pair as judged by sequential single step photobleaching of both dyes were selected for FRET analysis (see Materials and methods and *Figure 5—figure supplement 1* for details). FRET efficiency was calculated for SecY**EG alone or with SecA and various nucleotides and plotted as histograms (*Figure 5B-G* and *Figure 5—figure supplement 2A*).

In the absence of SecA or nucleotide, SecY**EG exists mainly as a single population with an average FRET efficiency ($E_{FRET}$) of 0.76 (*Figure 5B*). Such high efficiency is expected for the closed state structure (*Figure 4A*). In accord with the ensemble data, addition of SecA causes the open states to become more populated, with the relative populations dependent on which nucleotide is provided (*Figure 5C–G*). While a two state model could fit the majority of the distributions (data not shown) both global non-linear fitting (*Figure 5—figure supplement 3A*) and singular value decomposition (*Figure 5—figure supplement 3B*) indicate that at least three components are required to fully describe the histogram distribution with ATP. Therefore three principal states were used to fit the entire dataset: (1) closed, $E_{FRET}$ ~0.76 (red); (2) part-open, $E_{FRET}$ ~0.59 (green); and (3) open, $E_{FRET}$ ~0.45 (blue).

Estimating absolute distances from $E_{FRET}$ values is not warranted in such a complex system, particularly as the dyes are attached by long, flexible linkers. However, anisotropy measurements indicate that the dyes retain rotational freedom in all states (*Figure 5—figure supplement 2C*). Hence, relative movements can be approximated from changes in $E_{FRET}$. Assuming an $R_0$ of 6 nm, the difference between the open and closed state is ~0.9 nm – consistent with the changes observed across LG by MD (*Figure 2C*) and those expected from the crystal structures (*Figure 4A*).

After adjusting for ~50% of SecY**EG oriented such that it cannot bind SecA (*Figure 4—figure supplements 1* and *2*) the relative percentages of closed, open and part-open can be determined for active SecYEG. These data (*Figure 5H*) show that in the presence of AMPPNP – which traps the complex in an ATP bound-like state—the LG occupies mainly the open conformation (*Figure 5C+H*), while with ADP the closed state is more highly populated (*Figure 5D+H*). Intriguingly, in the presence of ATP—which under steady-state conditions is rapidly hydrolysed to ADP (*Robson et al., 2009*)—SecY predominantly populates the part-open state (*Figure 5E+H*). Mathematical modelling indicates that the third state cannot be explained by averaging of the two different nucleotide bound forms during data collection (*Figure 5—figure supplement 3C+D*). So, the part-open state most likely represents a true intermediate on the pathway between the closed and open conformations.

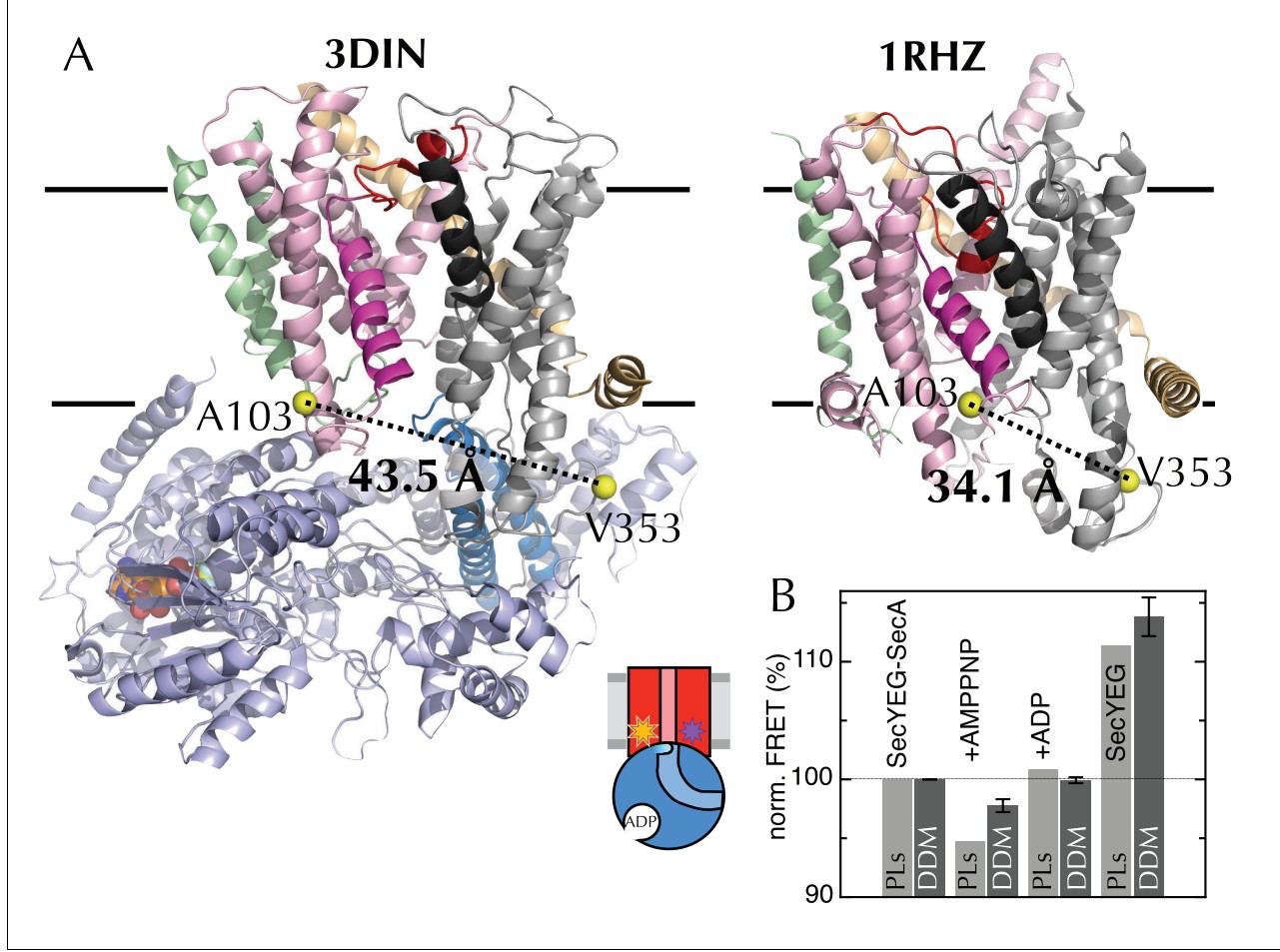

**Figure 4.** The extrinsic FRET pair of SecY**EG reports on the distance across the lateral gate. (**A**) Structures of SecYEG-SecA from *T. maritima* (left; part-open; PDB code 3DIN [***Zimmer et al., 2008***]) and SecYEβ from *M. jannaschii* (right; closed; PDB code 1RHZ [***Van den Berg et al., 2004***]). TMs 1–5 of SecY are coloured pink (TM2 highlighted magenta), TMs 6–10 grey (TM7 highlighted black), SecE orange, SecG/β green and SecA pale blue (2HF highlighted in bright blue). Residues equivalent to A103 and V353 in *E. coli* (K103 and I342 in *T. maritima*; I94 and I356 in *M. jannaschii*) are shown as yellow spheres, with the distances between them (Cβ-Cβ) marked out: 43.5 Å in the open complex, and 34.1 Å in the closed complex. (**B**) FRET efficiencies of 100 nM SecY**EG in PLs (light grey) or DDM-solubilised (dark grey): with 1 µM SecA; with SecA and 1 mM AMPPNP; with SecA and 1 mM ADP; and alone. Data are normalised to SecYEG with SecA but without added nucleotide, and error bars represent the standard error from six repeats.

The following figure supplements are available for figure 4:

**Figure supplement 1.** The extrinsic FRET pair of SecY**EG reports on the distance across the lateral gate.

**Figure supplement 2.** Interpreting ensemble FRET in PLs.

Addition of pOA + ATP to SecY**EG-SecA causes the open state to dominate (***Figure 5F+H***); an effect consolidated by addition of a large excess of SecA and pOA (***Figure 5G+H***). This is consistent with the known properties of the steady-state cycle of SecA: ADP release is strongly accelerated by the presence of pre-protein, such that the open (ATP-bound) state becomes more highly populated (***Robson et al., 2009***). No further states of the channel are detected in the presence of the pre-protein (***Figure 5F+G***). Taken together, the MD and FRET results show that nucleotide occupancy in SecA regulates the opening of the SecY protein-channel and LG.

## Closure of the SecY lateral gate alters ATP turnover by SecA

In addition to the LG movement, the MD simulations showed differences in the NBS of SecA, which turns out to be more open in the ATP simulations compared to those done with ADP bound

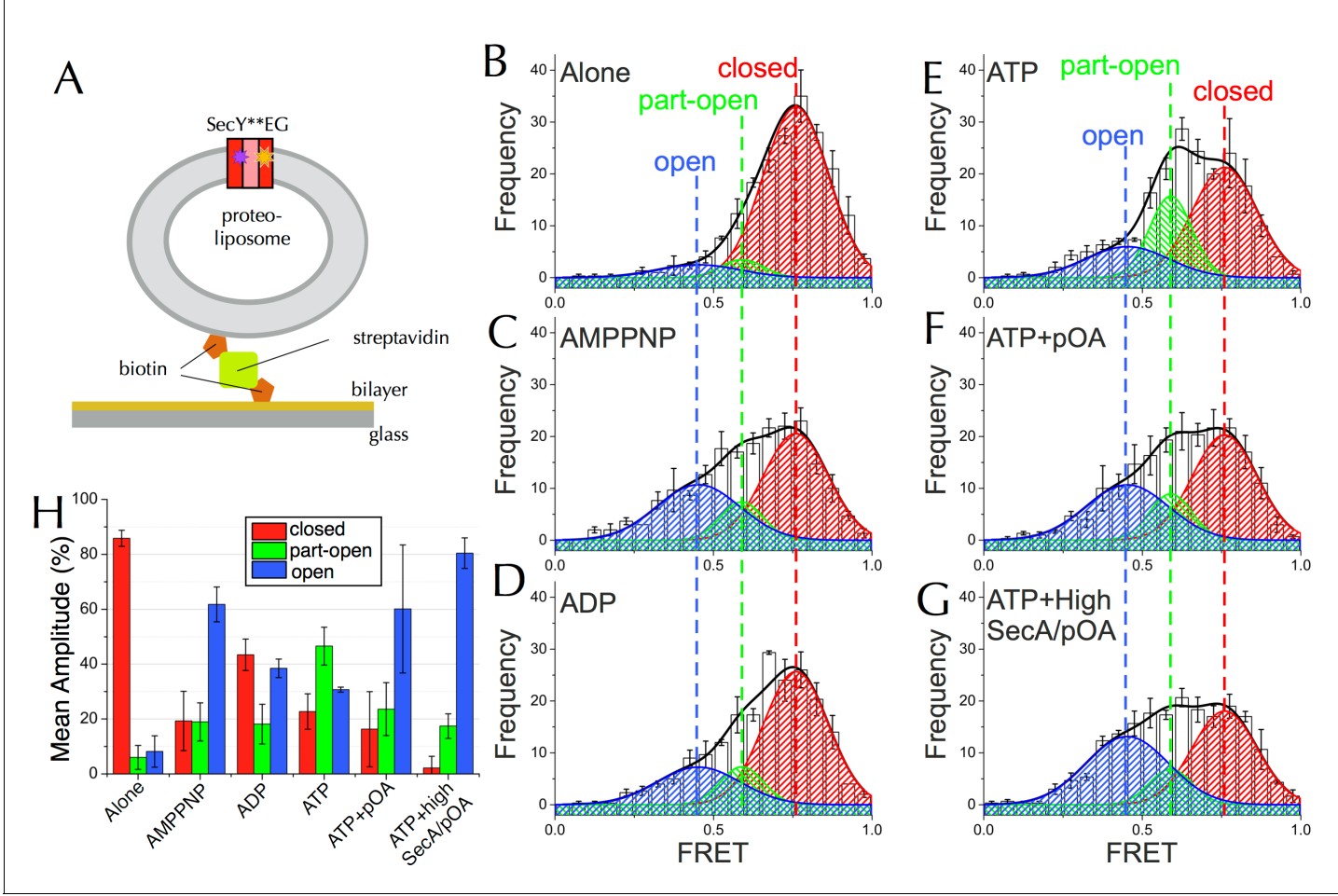

**Figure 5.** Communication from SecA to SecY: single molecule FRET analysis of lateral gate opening. (**A**) Schematic diagram of proteo-liposomes (PLs) containing a single SecY**EG complex used for single molecule TIRF experiments. PLs extruded to 100 nm were immobilised on the surface of a glass-supported lipid bilayer *via* a biotin-streptavidin-biotin bridge. (**B–G**) TIRF FRET efficiency distributions of SecY**EG PLs imaged alone (**B**) or incubated at room temperature for 15–40 min with 40 nM SecA in the presence of (**C**) 1 mM AMPPNP; (**D**) 1 mM ADP; (**E**) 1 mM ATP and an ATP regenerating system or (**F**) 1 mM ATP, 200 nM pOA and an ATP regenerating system, as well as (**G**) with 1 µM SecA, 1 mM ATP, 700 nM pOA and an ATP regenerating system. Each data set shows the average and SEM from three independent experiments (number of FRET events n = 200 each). Grey boxes represent histogram frequencies; red, green and blue shaded areas show least-squared Gaussian fits to high, medium and low FRET efficiencies, respectively. The black curve represents the sum of the fitted Gaussian distributions. (**H**) Amplitudes of the three TIRF FRET Gaussian peaks in panels (**B–G**). To correct for the inward-facing SecYEG molecules in the PLs (*Figure 4—figure supplement 1D*), 50% of the 'Alone' populations were subtracted from each sample (uncorrected data shown in *Figure 5—figure supplement 2B*). Bar heights are the mean from three replicates, and error bars represent the SEM computed by ANOVA analysis (full data shown in *Figure 5—figure supplement 2A*).

The following figure supplements are available for figure 5:

**Figure supplement 1.** Methodology for TIRF analysis.

**Figure supplement 2.** Complete single molecule FRET data.

**Figure supplement 3.** Three states are required to fit the smFRET data collected with ATP.

(*Figure 6A+B*). This can be quantified by calculating the nucleotide solvent accessible surface (*Figure 6—figure supplement 1A*; calculated with the GROMACS utility g_sas [*Eisenberg and McLachlan, 1986*]). It seems likely that an open NBS would increase the rate of nucleotide exchange, altering its affinity and turnover.

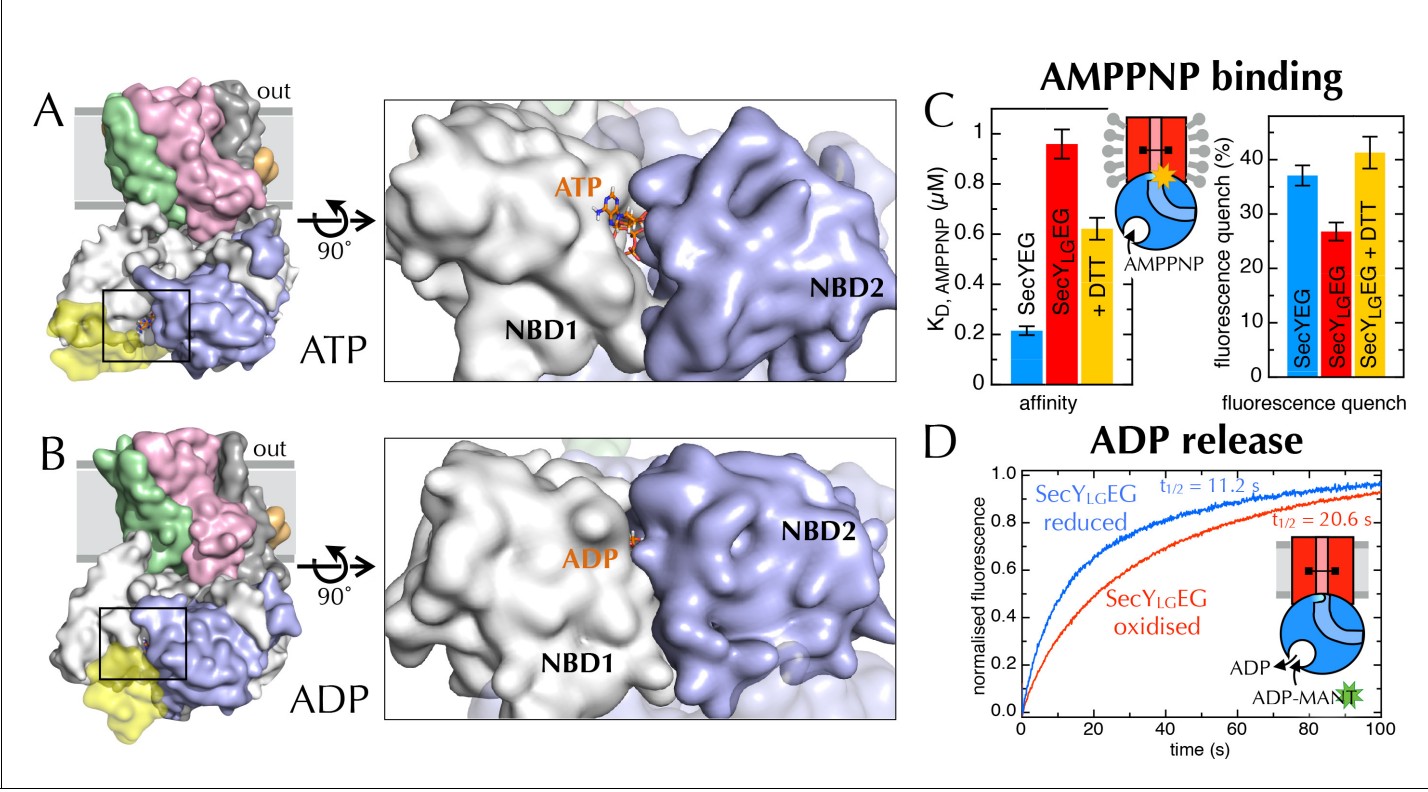

**Figure 6.** Communication from SecY to SecA: lateral gate-dependent changes in the NBS of SecA. (**A**) Surface view of SecYEG-SecA after 1 μs MD with ATP. SecY is coloured pink, SecE orange, SecG green and SecA white, with NBD2 highlighted in blue, and the domain exclusively found in thermophilic organisms coloured yellow. ATP is shown as sticks and coloured orange, with nitrogens blue, oxygens red and hydrogens white. In the closeup of the NBS (right), the yellow loop on SecA is omitted for clarity. (**B**) Same as panel **A**, but after MD with ADP. (**C**) Affinity for AMPPNP (left) and maximum fluorescence quenching (right) for SecA* in the presence of SecYEG (blue), SecY$_{LG}$EG (red), or SecY$_{LG}$EG with 100 mM DTT (yellow). AMPPNP was titrated into 200 nM SecA* in the presence of saturating (1 μM) SecYEG and fitted to the tight binding equation (see methods); error bars represent the SEM of three repeats. Note that 100 mM DTT is enough to fully reduce the disulphide bond (**Figure 6—figure supplement 2C**), and restore the fluorescence amplitude; the incomplete restoration of AMPPNP affinity is therefore most likely an effect of the cysteine substitutions. Note also a minor reduction in the affinity of SecY$_{LG}$EG for SecA in detergent (**Figure 6—figure supplement 3A**)—normally characteristic of the ADP-bound state (**Deville et al., 2011**). (**D**) Stopped flow fluorescence time courses of ADP release, after equilibrating 0.6 μM SecA and 1 mM ADP with either 2.4 μM SecYEG (blue) or SecY$_{LG}$EG (red) in PLs. Time courses were measured by rapidly mixing with 25 μM MANT-ADP and following fluorescence with excitation at 296 nm and emission measured using a 399 nm longpass filter. Data were fitted to double exponentials and normalised to give a total amplitude of 1.

The following figure supplements are available for figure 6:

**Figure supplement 1.** The nucleotide binding site in MD simulations.

**Figure supplement 2.** Validation of SecY$_{LG}$EG

**Figure supplement 3.** Additional data for the crosslinked SecY$_{LG}$EG.

A detailed examination of the MD data hints at a correlation between the width across the LG and accessibility of the NBS (**Figure 6—figure supplement 1B**), which suggests a more fundamental coupling between the two distant sites. If this communication is two-way—*i.e.* changes in SecY cause the NBS to open or close—it could provide a mechanism of coupling protein translocation to nucleotide exchange. To explore this possibility, a cysteine pair was introduced across the LG such that disulphide bond formation traps the translocon in the closed state (**du Plessis et al., 2009**) (SecY$_{LG}$EG; **Figure 6—figure supplement 2A**). Analysis of this sample by trypsin cleavage using SDS-PAGE shows >95% cross-linking (**Figure 6—figure supplement 2B+C**), and as reported previously

(*du Plessis et al., 2009*) the trapped complex is incapable of protein translocation (*Figure 6—figure supplement 2D*).

Previously, we have shown that the fluorescence of a fluorescein label on the tip of the 2HF (position 795 of *E. coli* SecA; hereafter SecA*) is strongly quenched upon association with SecYEG, but only in the ATP (AMPPNP) bound state, and not with ADP (*Deville et al., 2011*). We exploited this by titrating AMPPNP into SecYEG-SecA* or SecY$_{LG}$EG-SecA* to measure their relative affinities. Trapping the LG in a closed position causes a 5-fold reduction in affinity of SecA for AMPPNP (*Figure 6C*, left), demonstrating that the conformational state of the LG is transmitted to the NBS. The degree of quenching is also reduced, from 36% to 26% (*Figure 6C*, right and *Figure 6—figure supplement 3A*), hinting that the channel entrance occupies a more 'ADP-like' conformation.

The rate-limiting step during ATP turnover is the dissociation of ADP (*Robson et al., 2009*), so we also examined the effects of LG closure on the rate of ADP release from SecA ($k_{off}$). This can be measured by following competitive binding of a fluorescent nucleotide analogue (MANT-ADP) in a stopped flow apparatus (*Robson et al., 2009*). We found that the $k_{off}$ is slowed by ~50% with a cross-linked LG (*Figure 6D*), consistent with the results presented above showing that NBS and LG closure are coordinated. As expected, this cross-talk is also reflected in the steady-state ATP turnover parameters (*Figure 6—figure supplement 3B*). Together, the results establish the existence of a two-way communication between SecYEG and SecA over a distance of >5 nm: SecA ATP binding and hydrolysis regulate the aperture of the channel, while LG opening and closing controls the ATP affinity and rate of ADP release from SecA. Interestingly, binding of signal sequence into the LG is known to be an allosteric activator of SecA (*Gouridis et al., 2009*; *Hizlan et al., 2012*): the increase in the turnover of ATP caused by LG opening could provide a basis for this effect.

## Substrates are sensed at the entrance of the protein channel – a role for the two-helix finger?

The 2HF of SecA is situated directly on the path of pre-proteins as they enter the SecY-channel (*Figures 1B* and *7A+B*), and we have previously shown that perturbing the 2HF by cross-linking to SecY (A795C of SecA cross-linked to K268C of SecY; hereafter SecY$_x$EG-SecA$_{2HF}$; yellow dotted line in *Figure 7B*) stimulates the ATPase activity of SecA ~25-fold in PLs (*Whitehouse et al., 2012*). At the time, this observation was surprising; however, in light of the results above, we hypothesised that the 2HF might be the sensor that couples the movements of the LG and the NBS. Indeed, the 2HF is known to play a crucial role in coupling ATP hydrolysis to translocation (*Erlandson et al., 2008*).

To probe the importance of the 2HF, we first confirmed that the increase in ATPase activity is also reflected in an increased $k_{off}$ for ADP (using stopped flow as above; *Figure 7C*; ATPase increase in detergent shown in *Figure 7—figure supplement 1*). Next, to mimic a translocating pre-protein more directly, we introduced a cysteine at position 793 of *E. coli* SecA, *i.e.* on the 2HF (*Figure 7A +B*, orange sphere) and cross-linked a range of penta-peptides to it. The peptides were chosen to have different sequences (containing small or bulky side-chains) with a central cysteine for disulphide bond formation to the single engineered cysteine on SecA. A high degree of cross-linking was confirmed for the most hydrophobic substrate, AWCWA, using reverse-phase HPLC in the presence or absence of DTT (*Figure 7—figure supplement 2A*).

While the cysteine cross-linking site of SecA itself has no effect on the SecYEG stimulated release of ADP (*Figure 7D*, compare 'wt + SecYEG ' to 'G793C + SecYEG'), covalently cross-linked peptides elicit a significant increase in ADP $k_{off}$ (*Figure 7D* and *Figure 7—figure supplement 2B*). The bulkier peptide, AWCWA, has a greater effect than peptides with smaller residues (AGCGA or AFCFA). A similar effect was observed when the same peptides were cross-linked to the opposite side of the putative peptide binding path (on the helical scaffold domain (HSD); position 644 of *E. coli* SecA; *Figure 7A,B+D* and *Figure 7—figure supplement 2B*), but not at an unrelated position (position 20 of *E. coli* SecA; *Figure 7A* and *Figure 7—figure supplement 2B+C*).

Thus, we conclude that peptides are sensed at the SecY channel entrance, and that this information is transmitted back to the NBS. As both pulling the 2HF away from the channel (with a disulphide bond) and pushing it away (with a bulky peptide; *Figure 7B*) elicit an increase in ADP release, it seems plausible that the 2HF is sensing the passing peptide sterically; the dramatic effects of point mutations in the 2HF (*Erlandson et al., 2008*; *Bauer et al., 2014*) are consistent with this interpretation.

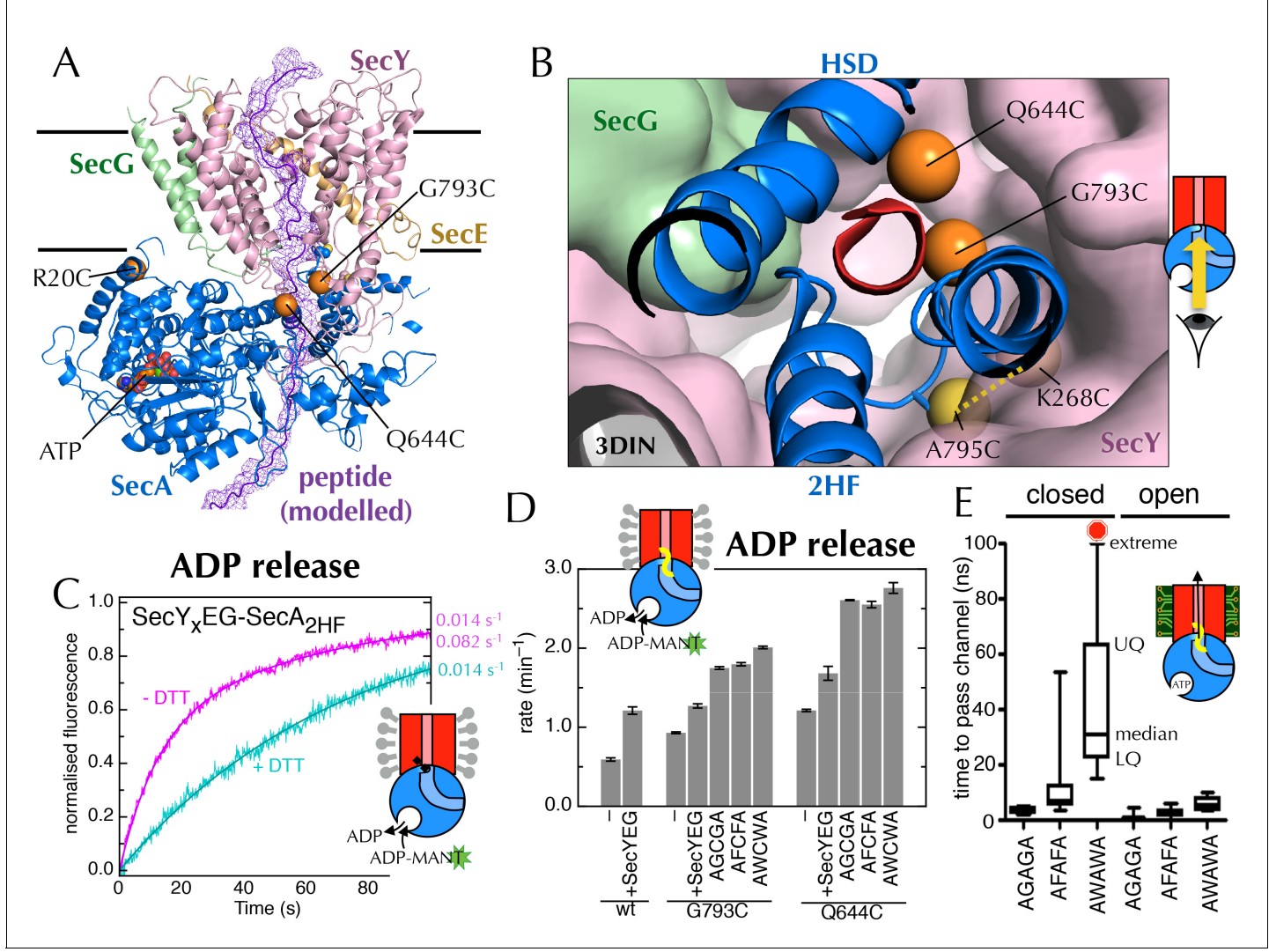

**Figure 7.** Involvement of the 2-helix finger in SecY <–> SecA communication.  (A) Overview cartoon structure of the 1 μs open SecYEG-SecA complex with SecY in pink, SecE orange, SecG green and SecA blue. Residues 27–76 of *E. coli* pOA, shown as a purple ribbon with a mesh surface, have been modelled into a likely pathway through the channel, based on known crosslinking data through SecA (***Bauer and Rapoport, 2009***) and SecY (***Cannon et al., 2005***). The three peptide crosslinking sites on SecA (R20C, Q644C and G793C in *E. coli* numbering; K32C, T640C and S780C in *T. maritima*) are shown with the Sγ atoms as orange spheres. (B) Closeup structure of the SecY channel entrance with SecA bound (*T. maritima*, PDB code 3DIN [***Zimmer et al., 2008***]), looking into the channel from the cytosolic side and coloured as in panel A. SecY, SecE and SecG are shown as surfaces, while the 2HF and HSD (helical scaffold domain) of SecA are shown in cartoon representation. The SecA peptide cross-linking sites SecA[644] and SecA[793] in *E. coli* (respectively SecA[640] and SecA[780] in *T. maritima*) are indicated by orange spheres; while the ATPase-activating SecA-SecY cross-linking sites (***Whitehouse et al., 2012***) (residues SecA[795]/SecY[268] in *E. coli*, SecA[782]/SecY[264] in *T. maritima*) are yellow spheres. To give an impression of size a short α-helix (red) has been modelled into the channel opening. (C) Stopped flow fluorescence traces of ADP release by 0.6 μM SecY$_x$EG-SecA$_{2HF}$ and 1 mM ADP without (magenta) or with (turquoise) 30 min pre-incubation with 100 mM DTT. Data were fitted to a double (magenta) or single (turquoise) exponential as described in methods. (D) ADP release rates determined by fluorescence stopped flow for wild-type SecA (wt), SecA[G793C] and SecA[Q644C]. Rates are shown for SecA alone; with an excess of SecYEG; and, where relevant, crosslinked to the peptides AGCGA, AFCFA or AWCWA (and in the presence of saturating SecYEG). Error bars represent the SEM from three independent experiments. Stopped flow traces are shown in ***Figure 7—figure supplement 2B***. (E) Transit time for the peptides AGAGA, AFAFA or AWAWA through the closed (left) and open (right) SecY pore during constant force steered MD simulations. Each simulation was repeated ten times to produce the box plots. Two AWAWA peptides failed to cross the closed SecY channel within 100 ns (red octagon).

The following figure supplements are available for figure 7:

**Figure supplement 1.** ATP turnover by crosslinked SecY$_x$EG-SecA$_{2HF}$.

**Figure supplement 2.** Supporting data for the 2HF labelling.

*Figure 7 continued on next page*

*Figure 7 continued*

**Figure supplement 3.** Entrance to the protein channel of SecY

## The closed channel poses a barrier to the passage of bulkier regions of substrate

Previous studies of translocation by SecYEG-SecA have shown that the ease with which a stretch of polypeptide chain passes through the substrate channel depends on the nature of the amino acids in that stretch (*Sato et al., 1997*; *Bauer et al., 2014*). For example, poly-glycine regions diffuse freely and rapidly backwards and forth (*Bauer et al., 2014*), while charged amino acids, especially positively charged, are excluded (*Nouwen et al., 2009*; *Liang et al., 2012*). It has also been shown that some combinations of amino acids form secondary structural elements even before they exit the ribosome (*Hardesty and Kramer, 2001*; *Lu and Deutsch, 2005*), while others remain natively unfolded. Given that the aperture of the SecY channel is nucleotide-dependent, we thought it might form a choke point for diffusion of bulky or structured regions of a substrate sequence: in the ADP-bound state, only uncharged, unfolded stretches of polypeptide may pass; whereas the ATP-bound state is less restrictive (*Figure 7—figure supplement 3A*).

To test this idea, we carried out steered MD experiments. Short peptides corresponding to those used for the cross-linking experiments (above) were modelled into the post-simulation 'closed' and 'open' structures, at the SecY channel entrance (*Figure 7—figure supplement 3B*), and pulled though the channel with a constant directional force. As expected given the dimensions of the open channel (*Figure 7—figure supplement 3A*), all three substrates crossed without difficulty (*Figure 7E*). For the closed structure however, while AGAGA readily passes through the channel, the larger substrates – especially AWAWA – are substantially retarded (*Figure 7E*). This indicates that the closed conformation – favoured by ADP occupied SecA – can permit passage of narrow regions of substrate, but selects against more bulky stretches. To allow these bulky regions through, the channel would need to open – as occurs when ATP is bound.

## Discussion

Over a decade has passed since the structure of the SecY complex was first determined (*van den Berg et al., 2004*). However the mechanism of translocation—how ATP turnover by SecA drives unfolded pre-proteins across the membrane through SecY—is not yet understood. At present, the most favoured model posits that substrate is pushed through the channel by the 2HF, with ATP providing energy *via* a power stroke (*Erlandson et al., 2008*). This model has recently been updated to allow free diffusion of some stretches of less bulky residues (*Bauer et al., 2014*); however, as we have argued previously (*Collinson et al., 2015*), evidence for the power stroke itself is lacking. Indeed, it is not clear from the structure of SecYEG-SecA where the 2HF would move, and how it could selectively bind and move both hydrophobic and charged regions of substrate. Furthermore, as immobilisation of the 2HF inside the channel does not prevent translocation (*Whitehouse et al., 2012*), any 'power-stroke' conformational change would have to be extremely subtle.

Here, we have combined all-atom molecular dynamics simulations with single molecule FRET and ensemble biochemical assays to investigate the events that couple the ATPase cycle of SecA with the rest of the complex. Four key observations were made: (i) the SecY channel is predominantly open with ATP bound to SecA and closed with ADP; (ii) the aperture of the channel regulates nucleotide exchange in SecA; (iii) this regulation is sensitive to the size of substrates at the SecYEG channel entrance; and (iv) the closed channel provides a selective barrier against larger substrates.

Taken together, these observations are consistent with a mechanism for coupling ATP hydrolysis in SecA to protein translocation that does not require a power stroke. In our proposed model (*Figure 8* and *Movie 1*) the direction of random substrate diffusion (Brownian motion [*Simon et al., 1992*]) is biased by the action of the ATPase. In the absence of blockages, the polypeptide diffuses freely backwards and forwards through the closed (ADP-bound) channel (state (i) in *Figure 8*). When a region of polypeptide that cannot pass through reaches the channel entrance (state (ii) – block region as green circle), it triggers nucleotide exchange – probably *via* the 2HF (state (iii)). This causes

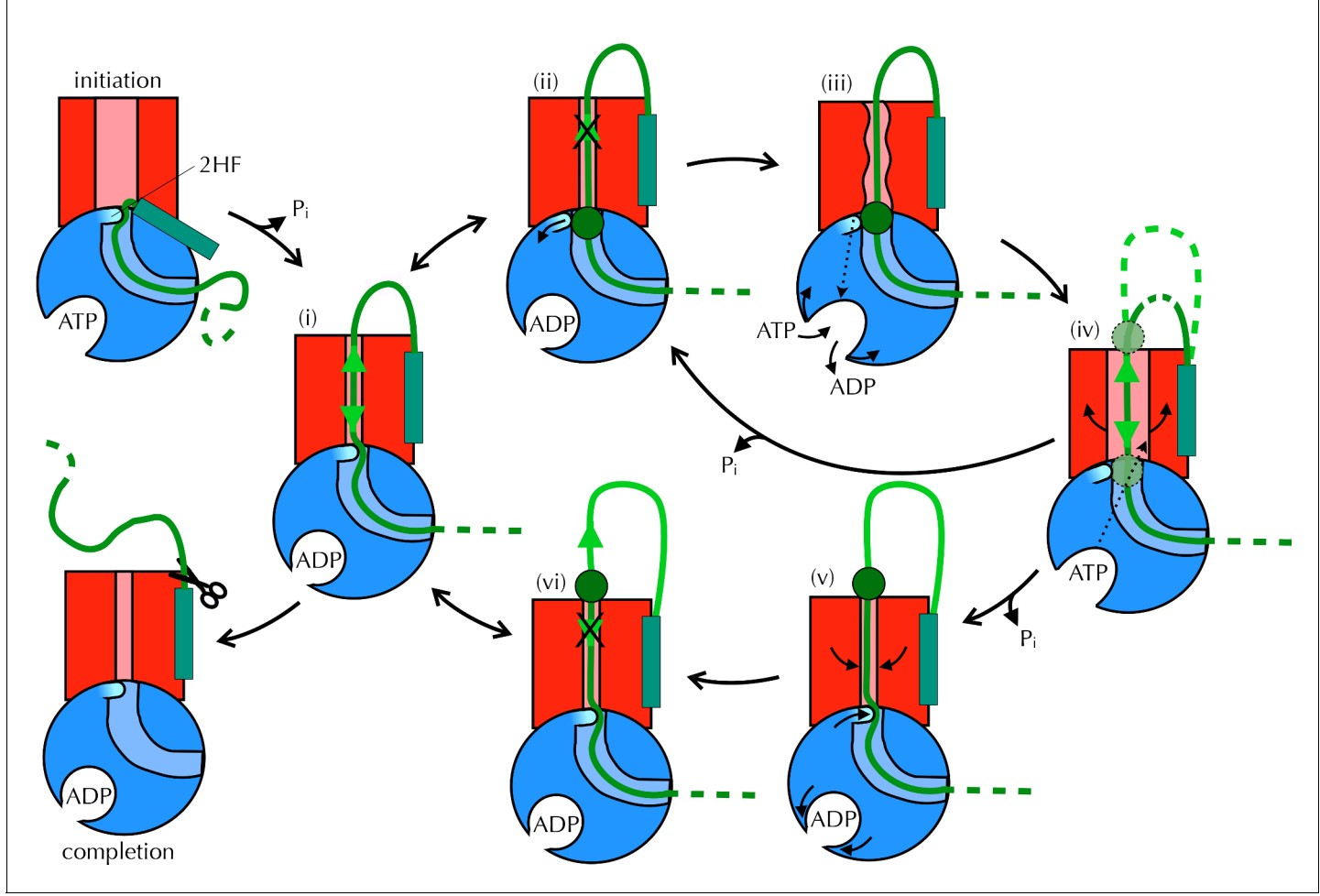

**Figure 8.** Proposed Brownian ratchet mechanism for translocation. SecYEG is shown in red (LG in light red), SecA in blue (substrate channel in light blue and 2HF in cyan and indicated), and substrate in green (with signal sequence as a turquoise rectangle). The initiation process (top left) involves conformational changes that prime the SecY-complex and intercalate the pre-protein (*Hizlan et al., 2012*; *Corey et al., 2016*). Subsequently, the substrate is free to diffuse backwards or forwards (i) until it reaches a block (green circle) at the entrance (ii) or exit (vi) to the channel. A block at the entrance triggers nucleotide exchange (iii), which leads to opening of the LG (iv). The wider channel permits diffusion of the blocked region of substrate within the pore, until ATP hydrolysis recloses the channel, trapping the block either outside (v) or inside (ii) the membrane. This cycle produces a net forwards driving force because blocks at the channel exit (vi) do not trigger nucleotide exchange and channel opening; the substrate is therefore ratcheted in one direction. Once the entire chain emerges from the channel it can no longer diffuse backwards and cleavage of the signal sequence is all that remains (completion).

a brief opening of the channel (iv), allowing the polypeptide to diffuse freely, before ATP is hydrolysed and the channel closes ((ii) or (v) depending on the position of the block). Back-diffusion is restricted because bulky regions on the periplasmic side do not trigger nucleotide exchange (vi): the whole scheme therefore acts as a ratchet promoting translocation. In essence, the bias for forward directionality arises because the energy transducing (ATP dependent) step – which resolves channel blockages – happens at the cytosolic, but not the periplasmic, surface. Moreover, the irreversibility of the initiation and completion steps would ensure forward directionality: once the signal sequence makes contact with the lipid bilayer (*McKnight et al., 1991*; *Hizlan et al., 2012*; *Corey et al., 2016*) it remains firmly fixed there, and is not cleaved off until most or all of the protein has crossed (*Josefsson and Randall, 1981*). This ensures that backward motion of the polypeptide does not lead to aborted transport.

It should be noted that we are limited by practical considerations in the size of peptide we can attach to the 2HF or the adjacent site on SecA. Nevertheless, its presence in the channel activates

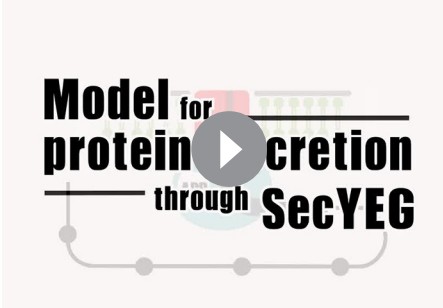

**Video 1.** Proposed Brownian ratchet mechanism for translocation The proposed Brownian ratchet model for protein translocation, as presented in *Figure 8*. Video produced by Nan Burston.

the NBS – the first demonstration of polypeptide-induced stimulation of the ATPase activity in solution (*i.e.* in the absence of a membrane). This selective response to differently sized peptides is very interesting: in the context of general protein transport, regions of folded secondary structure or stretches of polypeptide chain enriched in bulky amino acids are likely to occur (*Park et al., 2014*). These could be considerably larger than two tryptophans as in the peptide AWCWA. This, together with the fact that we see such variability in the width across LG with ATP bound (*Figure 2C*), suggests a degree of gradation in the mechanism: the larger and bulkier a region of substrate, the less likely it is to spontaneously cross the channel, but the more likely it is to induce nucleotide exchange. Indeed, at its core our model only requires a single open state to function: perhaps the most open state is reserved to conduct the largest substrates, after multiple rounds of ATP hydrolysis.

While previous models for translocation have postulated a substrate clamp within SecA, or large-scale conformational changes that physically push the substrate through the channel (*van der Wolk et al., 1997*; *Erlandson et al., 2008*; *Bauer et al., 2014*), the Brownian ratchet mechanism proposed here (*Figure 8* and *Video 1*) requires only the conformational changes predicted by MD and verified experimentally by ensemble and single molecule FRET studies. Moreover, because much of the peptide is diffusing rather than being pushed, the proposed model retains scope for further acceleration by the PMF (*Brundage et al., 1990*), folding, or by the binding of periplasmic chaperones: these factors would also prevent backsliding, and thereby promote flux towards the periplasmic side of the membrane.

The different open and closed states required for the model are represented in the various structures of the translocon determined in the presence or absence of translocation partners and engaged translocation substrate (*Van den Berg et al., 2004*; *Zimmer et al., 2008*; *Park et al., 2014*; *Gogala et al., 2014*; *Pfeffer et al., 2015*; *Voorhees and Hegde, 2016*). Indeed, a structure of SecYEG-SecA was very recently determined with a pseudo pre-protein within the channel (*Li et al., 2016*). With a narrow region of polypeptide (Ala-Gly-Gly) in the centre of the channel and ADP-BeF$_x$ bound to the NBS, the LG closely resembles that of 3DIN (*Zimmer et al., 2008*) – or the part-open conformation reported here. The pore-ring is packed tightly around this sequence, suggesting (i) that with substrate bound, the complex might not reach the fully closed state described here (upon ATP hydrolysis), and (ii) that further opening (upon ATP binding) would be required to accommodate a larger region of substrate – as we propose here.

It is important to point out that the model does not rule out further conformational changes that are not observable on the timescale of our MD simulation, or resolved by the single molecule FRET analysis presented. It has been reported that the ATP-bound state of SecA binds more tightly to substrate than the ADP-bound state (*Bauer et al., 2014*). This could be evidence of an additional pre-protein clamp in SecA (*Bauer and Rapoport, 2009*), perhaps designed to prevent the (iv) → (ii) transition in *Figure 8*: such a possibility is not precluded by the data, and would represent a refinement of the model presented here. Alternately, it could form part of a proof-reading mechanism to prevent poorly translocated substrates from clogging and jamming the Sec machinery (*Liang et al., 2012*). This is known to be deleterious to cells, leading to degradation of SecY (*van Stelten et al., 2009*); SecA has indeed been reported to have unfolding activity (*Arkowitz et al., 1993*).

The proposed model (*Figure 8* and *Video 1*) accounts for many previously noted properties of translocation: for example, the translocation intermediates sometimes observed at low ATP concentrations (*van der Wolk et al., 1997*) could be caused by regions of substrate with particularly high energy barriers to transit. It would also give rise to an approximately linear dependence of translocation rate on substrate length (*Liang et al., 2009*) – especially those with repeating sequences (*Tomkiewicz et al., 2006*) – while giving different rates for substrates with markedly different

sequences (*Sato et al., 1997*). Furthermore, the fact that ATP-bound SecA is bound tightly to the channel, while the ADP-bound form can be released (*Bauer et al., 2014*), makes sense from the perspective of cellular homeostasis and membrane integrity: release of SecA from a wide open SecY channel would allow the leakage of ions and compromise the energy conserving capabilities of the plasma membrane.

The above mechanism represents a breakthrough in our understanding of protein export through the ubiquitous Sec complex. It also resembles a suggested mechanism for the eukaryotic Sec machinery (*Matlack et al., 1999*), but with ratcheting from the cytosolic rather than the distal side of the membrane, which compensates for the absence of ATP in the periplasm. Regulated constriction of the channel, therefore, may offer a unifying concept for protein transport more generally, *e.g.* in mitochondrial and chloroplast protein import. Furthermore, two-way communication between the ATPase NBS and the polymer binding site over distances ranging from 2 to 15 nm has been observed for several molecular motors such as RecA-like helicases (*Mancini et al., 2004*; *Kainov et al., 2008*) and AAA+ dynein (*Kon et al., 2009*). Thus, a common mechanism is emerging that may underlie many other systems responsible for the conversion of chemical energy into directional motion of proteins and nucleic acids.

## Materials and methods

### Molecular dynamics simulations

All simulations were run in GROMACS 4.6.4 (*Berendsen et al., 1995*). Models for the simulations were built using chains A, C, D and E of the crystal structure 3DIN (*Zimmer et al., 2008*) as starting coordinates, with missing loops added using Modeller (*Sali and Blundell, 1993*) and ADP-BeF$_x$ replaced with either ADP or ATP (*Piggot et al., 2012*), or removed entirely. The protein was described using the OPLS-AA force field (*Jorgensen et al., 1996*) and embedded in a 512 united-atom POPC membrane (*Ulmschneider and Ulmschneider, 2009*), using the GROMACS utility g_membed (*Wolf et al., 2010*). Alternatively, the systems were described with Amber ff99SB-ILDN (*Lindorff-Larsen et al., 2010*) using a Lipid14 POPC membrane (*Dickson et al., 2014*). The protein-membrane structures were built into simulation boxes with periodic boundary conditions in all dimensions and solvated with explicit SPC water and sodium and chloride ions to a neutral charge and concentration of 0.15 M. In total there were approximately 230,000 atoms per simulation box. The systems were energy minimized using the steepest descents method over 2 x 5000 steps, then equilibrated for 1 ns using the NPT ensemble at 300 K with the Bussi-Donadio-Parrinello thermostat and semi-isotropic Parrinello-Rahman pressure coupling. Bond lengths were constrained using the LINCS method. Non-bonded interaction cut-offs were calculated using the Verlet method, with the neighbour search list updated every 20 steps. Long-range electrostatic interactions were calculated using the particle mesh Ewald method and a cut-off of 1.0 nm was applied for van der Waals and short range electrostatic interactions.

Molecular dynamics simulations were run with 2 fs integration time steps over 400–500 ns on the University of Bristol's High Performance Computer, BlueCrystal, and where applicable extended to 1 μs on the UK HPC facility ARCHER. All simulations reached a steady state as judged by their root-mean-squared-deviation from the starting structure (*Figure 2—figure supplement 1*).

For the steered MD 'pulling' experiments, a penta-peptide was modelled into a representative closed or open post-200 ns-simulation structure, in the cavity between SecA and SecY (*Figure 7—figure supplement 3B*). The penta-peptide was either AGAGA, AFAFA or AWAWA, with the backbone coordinates identical between the peptides. Simulations were run with the peptide in place for 10 ns to allow the region to relax, before a constant force of 400 kJ mol$^{-1}$ nm$^{-1}$ was applied to the peptide in a z-axis direction. The simulations were run until the peptide passed through the channel (or for 100 ns). Ten repeats were run for each, and the transit times were plotted using a box and whisker plot, showing extremes, upper and lower quartiles and median.

### Protein preparation

Site-directed mutagenesis was performed using the QuikChange protocol (Agilent) and confirmed by sequencing.

SecYEG, SecA, SecA$^{A795C-FL}$ (SecA*), SecY$_x$EG-SecA$_{2HF}$ and pOA were produced as described previously (**Gold et al., 2007**; **Deville et al., 2011**; **Whitehouse et al., 2012**). SecY**EG was produced in the same way as wild-type, then labelled for 45 mins on ice at 50 µM with 100 µM each of Alexa 488-C$_5$-maleimide and Alexa 594-C$_5$-maleimide (Invitrogen). The reactions were quenched with 10 mM DTT, and excess dye removed by gel filtration (Superdex-200, GE Healthcare, U.K.). Labelling efficiencies were between 75–90% for each dye, as determined using the manufacturer's quantification method and assuming a molar extinction coefficient of 70,820 cm$^{-1}$ for SecYEG. Membranes expressing SecY$_{LG}$EG, prepared as for wild-type, were oxidised with 1 mM copper phenanthroline prior to solubilisation in 1% n-Dodecyl β-D-Maltopyranoside (DDM). The samples were subjected to overnight batch-binding with Activated Thiol Sepharose 4B (GE Healthcare, U.K.), followed by centrifugation to remove uncross-linked protein. They were then treated by gel filtration as per the wild-type preparation.

SecA$^{G793C}$, SecA$^{Q644C}$ andSecA$^{R20C}$ were prepared as for wild-type. Peptides (AGCGA, AFCFA or AWCWA) were purchased from Cambridge Research Biochemicals, all N-terminally acetylated and C-terminally amidated. Cross-linking was performed by incubating each single cysteine variant with a 20-fold excess of peptide and 10 mM oxidised glutathione for 1 hr on ice. Excess free peptide was removed by gel filtration (Superdex-200). The efficacy of this method was confirmed for the AWCWA peptide using reverse-phase HPLC (0–50% acetonitrile in 0.1% TFA) on an XBridge BEH300 C4 column (Waters, U.K.), following absorbance at 280 nm (*Figure 7—figure supplement 2A*).

PLs of *E. coli* polar lipid containing SecYEG were produced as described previously (**Gold et al., 2007**).

## ATPase and translocation assays

ATPase activities were determined as described previously (**Gold et al., 2007**). In brief, ATP consumption was coupled to NADH depletion using a pyruvate kinase/lactate dehydrogenase regenerating system in the presence of excess phosphoenol pyruvate, and the absorbance at 340 nm followed. Rates were determined by fitting to a straight line, and the slopes (in $\Delta A_{340}.min-^1$) divided by the concentration of SecA and the molar extinction coefficient of NADH (6,220 M$-^1$ at 340 nm) to give (M ATP).(M SecA)$-^1$.min$-^1$. All data fitting for ensemble experiments was performed using pro Fit (Quansoft).

Translocation efficiencies were determined using an in vitro translocation assay (**Gold et al., 2007**): after a 40 min translocation reaction at 25°C, all untranslocated material was degraded with protease K, and the translocated material quantified by western blotting.

## Equilibrium fluorescence measurements

Fluorescence spectra of SecY**EG PLs were measured on a Jobin Yvon Fluorolog (Horiba Scientific), exciting at 493 nm and measuring emission spectra. Measurements were performed in TKM buffer (20 mM Tris pH 7.5, 50 mM KCl, 2 mM MgCl$_2$), starting with 50 nM SecY**EG, then adding sequentially 1 µM nucleotide-stripped SecA, 1 mM AMPPNP, ADP or ATP, and 0.7 µM pOA. At least 5 mins were left after each addition before measuring, and each spectrum was repeated three times to confirm that a steady state was reached. Data were corrected for dilution before plotting.

FRET measurements for SecY**EG in DDM were performed in TSG buffer (20 mM Tris pH 7.5, 130 mM NaCl, 10% glycerol) with 0.02% DDM and 2 mM MgCl$_2$ on a Nanodrop 3300 Fluorospectrometer (Thermo Scientific, Waltham MA, ). 100 nM DDM-solubilised SecY**EG was measured alone, with 1 µM SecA, with SecA and 1 mM ADP, or with SecA and 1 mM AMPPNP. After excitation with blue light, fluorescence signals at 519 nm (donor; F$_D$) and 617 nm (acceptor; F$_A$) were measured, and FRET efficiencies calculated as F$_A$ / (F$_D$ + F$_A$). For each data point, three repeats were taken and the average FRET efficiency used.

## Affinity assays

Affinity assays employing SecY**EG were performed by mixing 50 nM SecY**EG PLs with 500 µM ADP or AMPPNP in TKM buffer with varying amounts of SecA. Affinity assays reporting from SecA* were performed by mixing 10 nM SecA* in TSG buffer with 0.02% DDM, 2 mM MgCl$_2$, 1 mM AMPPNP and varying concentrations of SecYEG. AMPPNP affinity assays were carried out by mixing

varying concentrations of AMPPNP with pre-equilibrated 100 nM SecA and 500 nM SecYEG in the same buffer. DTT was either omitted or included at 100 mM; note that DTT has no effect on the interaction of SecA with wild-type SecYEG. Single channel fluorescence readings were taken at 522 nm and FRET readings at 519 nm and 617 nm, using a Nanodrop 3300 Fluorospectrometer with blue light excitation. The decreases in FRET or fluorescence were then plotted and fitted to a tight binding equation:

$$F = F_0 - B_{max} \frac{E_0 + s + K_D - \sqrt{(E_0 + s + K_D)^2 - 4E_0 s}}{2E_0}$$

where F is the fluorescence signal, $F_0$ is signal in the absence of substrate, $B_{max}$ is the amplitude of the fluorescence change, $E_0$ is the concentration of the fixed binding partner, s is the substrate concentration and $K_D$ is the dissociation constant.

## Single molecule FRET

SecY**EG was reconstituted into PLs with *E. coli* polar lipid to a final concentration of 1.5 nM, and extruded to 100 nm: at this concentration and size, most liposomes are expected to contain either 0 or 1 copy of SecY**EG (*Deville et al., 2011*). PLs were immobilised on a glass supported lipid bilayer and imaged with a TIRF microscope as described previously (*Figure 5A*). The buffer used was TKM with 1 mM 6-hydroxy-2,5,7,8-tetramethylchroman-2-carboxylic acid (TROLOX) and 71 mM β-mercaptoethanol as oxygen scavenging agents to extend florescent dye lifetimes. Immobilised samples were treated with 40 nM SecA and 1 mM ADP, 1 mM AMPPNP or 1 mM ATP and the ATP regeneration system used for ATPase assays, with or without 200 nM pOA. For the high SecA/high pOA sample (*Figure 5G*), 1 μM SecA and 700 nM pOA were used with 1 mM ATP and the ATP regeneration system. TIRF movies (200 ms frame rate) were taken from samples between 15 and 45 min after the addition of substrate.

The data processing flow is outlined in *Figure 5—figure supplement 1A*. The two channels of each image (*Figure 5—figure supplement 1B*) were aligned and fluorescence count traces (donor and acceptor) were extracted as described previously (*Sharma et al., 2014*) (*Figure 5—figure supplement 1C*). Since the system is essentially at equilibrium and the signals remain stationary we used the consecutive acceptor (first) and donor (second) single-step photobleaching events to identify complexes with exactly a single donor and single acceptor (*Figure 5—figure supplement 1D*). Only traces which exhibit an anti-correlated rise in donor signal upon acceptor photobleaching, a signature of single molecule FRET, are included in the analysis.

While this method discards FRET traces in which the donor photobleaches first, it eliminates the possibility of selecting traces with an ill-defined number of acceptor dyes in each immobilized PL. As the method provides access to the donor signal with acceptor present (before acceptor photobleaching) and absent (after acceptor photobleaching), it is a valid way to compute FRET efficiencies (*Lakowicz, 2013*) (*Figure 5—figure supplement 1D*). This method has the additional advantage that it does not need signal corrections for quantum yield changes or channel sensitivity. Note that while the acceptor signal is not directly used to calculate FRET efficiency its changes are employed to identify the smFRET traces and the relevant regions of the donor signal for data analysis (*Figure 5—figure supplement 1C*).

The experiments were repeated three times using independent PL preparations, then FRET values were collated as histograms (*Figure 5—figure supplement 2A*) and the three repeat histograms were averaged bin by bin to produce the final histogram with standard error for each bin (*Figure 5B–G*).

Singular value decomposition (SVD) of histograms for different conditions indicated than minimum of three components were necessary to describe the ATP data within experimental error (*Figure 5—figure supplement 3B*). Likewise, the 'ATP' individual repeat histograms can only be adequately decomposed into a minimum of three Gaussian components (*Figure 5—figure supplements 2* and *3A*). Using global fitting, the resulting amplitudes of the Gaussian peaks for the three independent repeats (*Figure 5—figure supplement 1A*) were subjected to ANOVA yielding mean amplitude and standard errors (SEM) (*Figure 5—figure supplement 2B*).

To measure dye anisotropy, 100 nM reconstituted SecY**EG was incubated for 5 min at 25°C alone or with 1 μM SecA and 1 mM ADP, AMPPNP or 1 mM ATP with the ATP regenerating system

and 700 nM pOA. A Florolog (Horiba Jobin Yvon) was use to collect emission spectra for dye anisotropy. The samples were either excited at 492 nm and 510–650 nm emission was detected or excited at 590 nm and 610–700 nm emission recorded (increment 2 nm, integration time 1 s, slits 5 nm). Sensitivity correction (G factor) was determined using excitation polarisation parallel to emission propagation direction. Using the G correction, anisotropy was computed and averaged over the peak of each emission band (515–525 and 615–625 nm).

### Kinetic simulations

Simulations of interconversion between the closed ($E_{FRET}$ = 0.76) and open states ($E_{FRET}$ =0.45) were done using Gillespie discrete event algorithm (*Gillespie, 1977*; *Gillespie, 1978*) with time step 1 ms and dwell times set at $1/k_{cat}$ ($k_{cat}$ = 0.27 s$^{-1}$, the rate of ADP release) for the high FRET and $1/k_{cleave}$ ($k_{cleave}$ = 11.5 s$^{-1}$ the rate of ATP hydrolysis and phosphate release) for the low FRET state, respectively (*Robson et al., 2009*). Normally distributed noise was added to match the width of the high FRET peak (0.24) and the data was then averaged over 200 ms windows to emulate sampling during TIRF data collection. The algorithm and subsequent averaging was implemented in Matlab.

### Limited proteolysis

For trypsin digest analysis, 4 µg SecYEG samples were incubated with 0.75 µg.ml$^{-1}$ (for proteins in DDM) or 6 µg.ml$^{-1}$ (for proteins in PLs) sequencing grade porcine trypsin (Promega, Southampton, U.K.) in a total volume of 10 µl at room temperature for 20 min, then analysed by SDS-PAGE. In the case of PLs, the reactions were quenched with 200 mM NaOH prior to gel analysis. Bands were visualised and quantified either using an Odyssey Fc (LI-COR Biosciences, Lincoln, NE; for Coomassie stained gels) or a Typhoon FLA 9500 (GE Healthcare) and ImageJ (for fluorescence scans).

### Stopped flow fluorescence measurements of ADP off rates

0.6 µM SecA with or without 2.4 µM SecYEG in either PLs (for experiments using SecY$_{LG}$EG) or 0.02% w/v DDM (wild-type SecYEG) were pre-mixed with 1 mM ADP to equilibrate and produce an ADP-bound state of SecA. This was mixed rapidly (<2 ms) with an equal volume of 25 µM MANT-ADP (Sigma Aldrich, St. Louis, MO) in the same buffer on a KinetAsyst SF-61SX2 stopped-flow (Hi-Tech). Tryptophan fluorescence was excited at 296 nm and emission was recorded using a 399 nm longpass filter; this allows monitoring of FRET between the tryptophans in SecA and the MANT moiety upon SecA binding. Because the off-rate for ADP << the on-rate for MANT-ADP, fitting to a single exponential yields the rate of ADP release (*Robson et al., 2009*).

Note that for experiments in PLs and with crosslinked SecY$_x$EG-SecA$_{2HF}$, a double exponential is required to fit the data. In the case of PLs (*Figure 6D*), this is likely a complication caused by the presence of lipids. For ease of comparison, we simply normalised the plots to give a total amplitude of 1 and rather than attempting to extract potentially spurious individual rates, we simply estimated the time taken to reach 50% completion ($t_{1/2}$). For SecY$_x$EG-SecA$_{2HF}$ (*Figure 7C*), we reasoned that SecY$_x$EG-SecA$_{2HF}$ is inevitably contaminated with some free SecA$^{A795C}$ (*Whitehouse et al., 2012*), so the populations most likely correspond to a mix of SecY$_x$EG-SecA$_{2HF}$ and SecA$^{A795C}$. We confirmed this by treating the complex with 100 mM DTT, which yields a single exponential rate of ADP release (*Figure 7C*, turquoise line). To fit the $k_{off}$ for SecY$_x$EG-SecA$_{2HF}$, we therefore fixed one exponential rate to that of the fully reduced complex, and allowed the other, along with both amplitudes, to float freely.

## Acknowledgements

This paper is dedicated to our friend and colleague, the late Prof. Steve Baldwin. Part of this work was carried out using the computational facilities of the Advanced Computing Research Centre, University of Bristol—http://www.bris.ac.uk/acrc/. The extended simulations were carried out with computer time on ARCHER provided by HECBioSim, the UK High End Computing Consortium for Biomolecular Simulation (hecbiosim.ac.uk), supported by the EPSRC. This work was funded by the BBSRC: IC and WJA (BB/I006737/1); RAC (BBSRC South West Bioscience Doctoral Training Partnership and BB/M003604/I) and SAB, SER, RT and PO (BB/I008675/1). Additional support was provided by the Wellcome Trust (104632) to IC and WJA; ERC ((FP7/2007-2013) / ERC grant

agreement 32240) to SER. We are extremely grateful to K Moreton and D Carter for their excellent technical support. We would also like to thank AR Clarke for critical reading of the manuscript.

## Additional information

### Funding

| Funder | Grant reference number | Author |
|---|---|---|
| Biotechnology and Biological Sciences Research Council | BB/I006737/1 | William John Allen Ian Collinson |
| Wellcome Trust | 104632 | William John Allen Ian Collinson |
| Engineering and Physical Sciences Research Council | HECBioSim | Robin Adam Corey Richard Barry Sessions |
| Biotechnology and Biological Sciences Research Council | BB/M003604/1 | Robin Adam Corey |
| Biotechnology and Biological Sciences Research Council | BB/I008675/1 | Peter Oatley Sheena E Radford Roman Tuma |
| European Research Council | FP7/2007-2013 | Sheena E Radford |

The funders had no role in study design, data collection and interpretation, or the decision to submit the work for publication.

### Author contributions

WJA, RAC, PO, Conception and design, Acquisition of data, Analysis and interpretation of data, Drafting or revising the article; RBS, SAB, SER, RT, Conception and design, Analysis and interpretation of data, Drafting or revising the article; IC, Conception and design, Analysis and interpretation of data, Drafting or revising the article

### Author ORCIDs

William John Allen, http://orcid.org/0000-0002-9513-4786
Sheena E Radford, http://orcid.org/0000-0002-3079-8039
Ian Collinson, http://orcid.org/0000-0002-3931-0503

## Additional files

### Major datasets

The following previously published datasets were used:

| Author(s) | Year | Dataset title | Dataset URL | Database, license, and accessibility information |
|---|---|---|---|---|
| van den Berg B, Clemons Jr WM, Collinson I, Modis Y, Hartmann E, Harrison SC, Rapoport TA | 2004 | The structure of a protein conducting channel | http://www.rcsb.org/pdb/explore/explore.do?structureId=1RHZ | Publicly available at the RCSB Protein Data Bank (accession no. 1RHZ) |
| Zimmer J, Nam Y, Rapoport TA | 2008 | Crystal structure of the protein-translocation complex formed by the SecY channel and the SecA ATPase | http://www.rcsb.org/pdb/explore/explore.do?structureId=3DIN | Publicly avaialble at the RCSB Protein Data Bank (accession no. 3DIN) |

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
