## [Decision Letter]

Thank you for submitting your article "Two-way communication between SecY and SecA suggests a Brownian ratchet mechanism for protein translocation" for consideration by *eLife*. Your article has been favorably evaluated by John Kuriyan (Senior editor) and three reviewers, one of whom is a member of our Board of Reviewing Editors.

The following individual involved in review of your submission has agreed to reveal his identity: Yibing Shan (peer reviewer).

The reviewers have discussed the reviews with one another and the Reviewing Editor has drafted this decision to help you prepare a revised submission.

Allen et al. have presented a combined computational and experimental analysis of protein translocation through the SecY-SecA translocation apparatus. The conclusion is that an interplay between pre-protein, nucleotide state of the SecA ATPase, and conformation of the SecY channel act to bias the directionality of pre-protein diffusion toward the periplasm. In this model, the bias arises from the fact that pre-protein bulkiness is preferentially detected (and subsequently relieved) at the cytosolic side of the channel. The key elements of this model are four-fold. First, MD simulations, bulk FRET, and single-molecule FRET studies suggest that widening of the lateral gate, and hence the pore diameter, of the SecY channel is favored by the ATP state of SecA. Second, a narrow SecY lateral gate enforced by a disulfide bridge can communicate to SecA to slow ADP release (and hence ATP binding). Third, steered MD simulations suggest that bulky peptides have more trouble crossing a narrow SecY pore than less bulky peptides. Fourth, bulky peptides crosslinked to the SecA two-helix finger at the cytosolic opening of the SecY channel communicate to hasten ADP release. When combined with earlier observations, this leads to a model where non-bulky stretches of peptide can move through the narrow SecY, bulky residues on the periplasmic side cannot backslide effectively through the channel, while bulky residues near the cytosolic side trigger channel widening to permit their diffusion through the channel.

The referees all agreed that the study is original, of high technical quality, and an important contribution to this field. The most salient referee comments are provided below. During the discussion leading from these comments, the following two points were felt to be essential in preparing a revised manuscript. The other referee comments are less central, but should be addressed with revisions to the text as necessary.

1) The interpretation of the experiment crosslinking peptides to the two-helix finger should be more cautious at this stage. It was felt that with the currently available data, the conclusion that bulky residues are 'sensed' by the two-helix finger was premature. Unless a strong justification can be provided, it is probably safer to conclude that positioning bulky peptides in the cytosolic vestibule of SecY stimulates ADP dissociation by a yet undermined mechanism. One can certainly present ideas in the Discussion about possible mechanisms, including sensing by the two-helix finger, but this should be clearly indicated as speculation.

2) Referee 3 raises an important point about the speed of conformational changes observed (on the nanosecond scale) relative to expectation from other systems. As this might imply the (interesting) possibility that the starting structure could represent a transition state, the referee recommends performing some additional relatively short (microsecond scale, or shorter – long enough to see the conformational transitions) simulations. In particular, this would involve verifying the result with a different force field, and additional simulations with the apo state. It is strongly recommended that this issue be addressed if at all possible.

In addition to these two issues, please note that two of the three referees also recommended a more complete discussion of the relationship of the new model to earlier model(s). This would seem important to add to the Discussion.

Reviewer #1:

Overall, I thought the paper presented a nice analysis that was clearly described, and provides a new and plausible mechanism for translocation. The topic is certainly of wide interest, and it is likely to stimulate further analysis in this field and may cause analogous processes in other ATP-driven systems to be re-considered. Thus, I generally support publication in *eLife*, pending the addressing of a few queries:

1) I feel the Discussion needs a more thorough comparison of the new model with the earlier "pushing" model to better explain to the reader why they favor a ratchet. In particular, the somewhat superficial resemblance of the idea that also placed a key emphasis on bulky residues (in that case, as the point of pushing) will leave readers wondering whether the earlier observations can now be reconciled into the new model. It is also worth discussing whether there was really any strong evidence for pushing per se (i.e., conformational changes that markedly move the 2HF) and whether such movement is needed in the new model.

2) The part of the argument that was least convincing to me was the sensing of bulky residues by the 2-helix finger. The authors directly crosslinked peptide to it, which seemed to me not a good way of determining if it is 'sensing' peptide since the putative 'sensor' is now perturbed. Would it not be better to choose a nearby site (and one further away as a negative control) to crosslink the peptides? This way, one can see if the 2HF is really able to detect peptides in its proximity. Alternatively, perhaps some other nearby region of SecA (or SecY) is doing the actual sensing. Or perhaps the authors are imagining a scenario where the extra space occupied by the bulkier peptide forces the 2HF into a different position. Either way, I can't quite see how crosslinking peptide to the 2HF is mimicking preprotein bulkiness, and a better explanation of what the authors think is going on here (and any alternatives that remain to be tested) is warranted.

3) In the model (i.e., the movie), the signal peptide is shown to remain translocon-bound until after translocation is complete. Is there evidence for this or is this just speculation? It is not an entirely trivial point because the new SecA-SecY structure with a putative signal peptide (Li et al., Nature, 2016) suggests that the signal binds between helices 2 and 7. Thus, if it stays there, the SecY might not be able to close in the ADP state (although probably beyond the scope of this paper, an MD simulation with the signal bound at this site would certainly be interesting). Some discussion of this new work and how it might relate to the model is warranted. If a bound signal essentially prevents lateral gate closing, it would seem that a key part of the model would need to be revised.

Reviewer #2:

The data presented in this manuscript is of a very high quality. The authors provide compelling evidence that the nucleotide-binding state SecA influences the conformation in SecYEG and that binding to ATP appears to promote an "open" conformation of SecYEG. The evidence suggesting that the presence of nascent substrate proteins in the channel influences the rate of nucleotide exchange by SecA is similarly convincing. The authors also suggest that the 2HF of SecA senses the presence of a preprotein in the channel and that this can be transmitted back to the nucleotide binding site. The interaction of preproteins with the 2HF is required for translocation, and the 2HF has therefore been implicated in physically pushing preproteins through SecYEG. Although it isn't clear from the data that the 2HF is the *only* feature of SecA that senses the presence of preprotein, the evidence clearly suggests that the 2HF is involved in sensing the presence of preprotein.

The Brownian-ratchet-type model proposed by the authors is consistent with the experimental data and could potentially provide an explanation for some of the inconsistencies with more deterministic models for SecA-driven translocation. Regardless of whether SecA-driven translocation is deterministic or stochastic, the experiments provide important insight into the mechanism of SecA-driven translocation. I feel that the manuscript merits publication in *eLife*. I have only a few comments. However, I do not feel any additional experiments are required for publication.

1) The stochastic model proposed by the authors is interesting, consistent with the data presented and potentially resolves a number of inconsistencies with previously proposed models. However, it isn't clear why the data presented in this paper is inconsistent with more deterministic models. For example, is it possible that this data is consistent with a variation of the "push-and-slide" model proposed by Bauer et al?

2) The authors do not deal consistently with the issue of back-sliding in the discussion of their Brownian-ratchet model. For example, in the second paragraph of the Discussion, the authors state, "Back diffusion is prevented because blocks at the exit do not trigger nucleotide exchange." This (rather obscure) statement appears to contradict a later suggestion in the fourth paragraph of the Discussion that a second (?) clamping mechanism by SecA could prevent backsliding.

3) In the second paragraph of the Discussion the authors suggest that binding of the signal sequence to SecYEG (or potentially partitioning of the signal sequence into the cytoplasmic membrane) ensures directionality. This insurance would require that the signal sequence be cleaved after the completion of translocation. However, multiple lines of evidence (e.g. Randall and colleagues. Cell 25:151) suggest that the signal sequence is cleaved off during the process of translocation. Wouldn't this make translocation reversible in their model?

Reviewer #3:

Regarding the molecular dynamics simulations of this study, I think an important opportunity would be missed if the investigators do not look into the simulation data deeper and try to understand--in atomic details the simulations afford – the differential local effects ADP- and ATP-binding give rise to and the allosteric mechanism propagating the effects to the pore. A basic biophysical understanding and the conformational changes in the nucleotide binding sites, presumably involving key conserved residues, should be discussed. (Potentially relevant, there are two analogous papers that show two distinct states ensues ADP-/ATP-binding of a protein kinase [Shan et al., PNAS, 2009; Foda et al., Nature Comm, 2015]).

Separately, I am concerned of the fastness of the pore closing/opening following placing ADP/ATP in the nucleotide binding site (~20 nanosecond). This is unusually, almost suspiciously fast, considering that typically such conformational changes occur at micro- to millisecond timescale and, because in SEC the information originating for the binding site has to travel nanometers to the pore, I would expect this to take significantly more than several microseconds. It is possible that the crystal structure by serendipity is positioned at a high-energy (possibly even the transition state, which would be very interesting and informative). If so, no barrier-crossing is required, which would be consistent with the nanosecond timescale observed. This should be further examined. Would we qualitatively see the same effect if a different force field (Amber or Charmm) is used? What happens if the nucleotide binding site is unoccupied in the simulations? (The new simulations do not need to run to 1 microsecond each if the conformational change occurs in 10s of nanoseconds).

---

## [Author Response]

1) The interpretation of the experiment crosslinking peptides to the two-helix finger should be more cautious at this stage. It was felt that with the currently available data, the conclusion that bulky residues are 'sensed' by the two-helix finger was premature. Unless a strong justification can be provided, it is probably safer to conclude that positioning bulky peptides in the cytosolic vestibule of SecY stimulates ADP dissociation by a yet undermined mechanism. One can certainly present ideas in the Discussion about possible mechanisms, including sensing by the two-helix finger, but this should be clearly indicated as speculation.

We agree – our interpretation that the 2HF senses peptide is speculation; although it seems likely given our data and previous work on the importance of the 2HF. We have amended our Results and Discussion to make this clearer. Note that we have included additional data that further supports the 2HF-sensor concept (see below).

2) Referee 3 raises an important point about the speed of conformational changes observed (on the nanosecond scale) relative to expectation from other systems. As this might imply the (interesting) possibility that the starting structure could represent a transition state, the referee recommends performing some additional relatively short (microsecond scale, or shorter – long enough to see the conformational transitions) simulations. In particular, this would involve verifying the result with a different force field, and additional simulations with the apo state. It is strongly recommended that this issue be addressed if at all possible.

Nanosecond timescale conformational changes across the distances observed here (>5 nm) are indeed unusual. The referee is probably correct that the input structure represents a high energy intermediate on the ATPase cycle. As stated in the manuscript (now clarified further), ADP-BeF_x_ has previously been reported as producing transition state-like intermediates. We believe this is what has happened here. Indeed, in previous work, we have shown that when solution hydrogen deuterium exchange experiments are performed (by mass spec), the results more closely resemble those of MD-relaxed structures than the original crystal coordinates (Radou et al. 2014). However, this does not prohibit same starting structure from sampling different conformations, depending on which nucleotide is bound.

With this in mind, the rapid relaxation to the lower energy ADP-bound state (with no energy barrier to cross) is unsurprising. This observation is supported by simulation data collected where the system was run with ATP present, which was then ‘hydrolysed’ to ADP. In this case, significant channel closing was not seen within 100 ns of simulation data (data not included), implying the actual speed of conformational changes is slower.

We have set up and run simulations in apo conditions, as per the referee’s suggestion (two repeats, Figure 9). These show a rapid closing of the LG, but to a lesser degree than for ADP.10.7554/eLife.15598.025Author Response Image 1.**DOI:**
http://dx.doi.org/10.7554/eLife.15598.025

This supports the idea that the input structure is not at an energy minimum. However, the exact conformational change is unlikely to have direct physiological relevance: the apo state is transient and would not normally follow the ATP-bound state. As such, we prefer not to include this data in the paper itself.

To further address the referee’s concerns, and provide evidence that the conformational changes are not simply due to force field bias, we have carried out additional simulations using the Amber ff99SB-ILDN force field with a Lipid14 POPC membrane. The results of this work are similar to those obtained using OPLS-AA, and are shown in a new supplemental figure (Figure 2—figure supplement 2).

In addition to these two issues, please note that two of the three referees also recommended a more complete discussion of the relationship of the new model to earlier model(s). This would seem important to add to the Discussion.

We have added more detail of previous models at the beginning of the Discussion.

Reviewer #1:

Overall, I thought the paper presented a nice analysis that was clearly described, and provides a new and plausible mechanism for translocation. The topic is certainly of wide interest, and it is likely to stimulate further analysis in this field and may cause analogous processes in other ATP-driven systems to be re-considered. Thus, I generally support publication in eLife, pending the addressing of a few queries:

1) I feel the Discussion needs a more thorough comparison of the new model with the earlier "pushing" model to better explain to the reader why they favor a ratchet. In particular, the somewhat superficial resemblance of the idea that also placed a key emphasis on bulky residues (in that case, as the point of pushing) will leave readers wondering whether the earlier observations can now be reconciled into the new model. It is also worth discussing whether there was really any strong evidence for pushing per se (i.e., conformational changes that markedly move the 2HF) and whether such movement is needed in the new model.

"Pushing" by the 2HF was inferred from the observation that the 2HF is protected from proteolysis in the presence of SecYEG and AMPPNP but not ADP. However, this is not necessarily evidence of pushing: the data can also be explained by the higher affinity of SecA for SecYEG when AMPPNP is bound. In our opinion, therefore, there is no direct evidence for a pushing role of the 2HF.

The model we present does not require any such conformational change. Indeed, we think this is one of its strengths: crosslinking the 2HF into the SecY channel does not prevent translocation, contrary to its proposed action as a piston (see (Whitehouse et al. 2012)).

We have amended the Discussion to reflect this point.

2) The part of the argument that was least convincing to me was the sensing of bulky residues by the 2-helix finger. The authors directly crosslinked peptide to it, which seemed to me not a good way of determining if it is 'sensing' peptide since the putative 'sensor' is now perturbed. Would it not be better to choose a nearby site (and one further away as a negative control) to crosslink the peptides? This way, one can see if the 2HF is really able to detect peptides in its proximity. Alternatively, perhaps some other nearby region of SecA (or SecY) is doing the actual sensing. Or perhaps the authors are imagining a scenario where the extra space occupied by the bulkier peptide forces the 2HF into a different position. Either way, I can't quite see how crosslinking peptide to the 2HF is mimicking preprotein bulkiness, and a better explanation of what the authors think is going on here (and any alternatives that remain to be tested) is warranted.

As suggested, we have now crosslinked the same peptides to the opposite side of the pre-protein channel, which produced a similar effect. In addition, we crosslinked the bulky (AWCWA) peptide to a position distant from the channel and observed no effect on ADP release. These results have been incorporated into the paper (Figure 7, Figure 7—figure supplement 2 and Figure 7—figure supplement 2), and the three crosslinking positions have been mapped onto the structure (Figure 7).

We have also added more detail of what we think is happening, and reworded to make clear what is shown experimentally and what is speculation.

3) In the model (i.e., the movie), the signal peptide is shown to remain translocon-bound until after translocation is complete. Is there evidence for this or is this just speculation? It is not an entirely trivial point because the new SecA-SecY structure with a putative signal peptide (Li et al., Nature, 2016) suggests that the signal binds between helices 2 and 7. Thus, if it stays there, the SecY might not be able to close in the ADP state (although probably beyond the scope of this paper, an MD simulation with the signal bound at this site would certainly be interesting). Some discussion of this new work and how it might relate to the model is warranted. If a bound signal essentially prevents lateral gate closing, it would seem that a key part of the model would need to be revised.

The signal sequence is thought to remain associated for most or all of the translocation event. We have added reference to this to the Discussion.

The conformation of the channel in the Li et al. resembles our 'part open' conformation, with a narrow polypeptide in the channel (although no 2HF). As passage of a wider substrate would probably require further opening, this seems to support the model. We have extended the Discussion to incorporate this new information.

MD with substrate is indeed beyond the scope of this paper. However, it would be useful and informative and we aim to perform this as part of a follow up paper.

Reviewer #2:

The data presented in this manuscript is of a very high quality. The authors provide compelling evidence that the nucleotide-binding state SecA influences the conformation in SecYEG and that binding to ATP appears to promote an "open" conformation of SecYEG. The evidence suggesting that the presence of nascent substrate proteins in the channel influences the rate of nucleotide exchange by SecA is similarly convincing. The authors also suggest that the 2HF of SecA senses the presence of a preprotein in the channel and that this can be transmitted back to the nucleotide binding site. The interaction of preproteins with the 2HF is required for translocation, and the 2HF has therefore been implicated in physically pushing preproteins through SecYEG. Although it isn't clear from the data that the 2HF is the only feature of SecA that senses the presence of preprotein, the evidence clearly suggests that the 2HF is involved in sensing the presence of preprotein.

The Brownian-ratchet-type model proposed by the authors is consistent with the experimental data and could potentially provide an explanation for some of the inconsistencies with more deterministic models for SecA-driven translocation. Regardless of whether SecA-driven translocation is deterministic or stochastic, the experiments provide important insight into the mechanism of SecA-driven translocation. I feel that the manuscript merits publication in eLife. I have only a few comments. However, I do not feel any additional experiments are required for publication.

1) The stochastic model proposed by the authors is interesting, consistent with the data presented and potentially resolves a number of inconsistencies with previously proposed models. However, it isn't clear why the data presented in this paper is inconsistent with more deterministic models. For example, is it possible that this data is consistent with a variation of the "push-and-slide" model proposed by Bauer et al?

We have expanded the Discussion to add more detail about previous models.

In essence, the 'push-and-slide' model paper provides very good evidence for the 'slide' aspect, but none for the 'push', which is assumed. So while the data presented here do not prohibit a power stroke, they obviate the need for one to explain translocation.

2) The authors do not deal consistently with the issue of back-sliding in the discussion of their Brownian-ratchet model. For example, in the second paragraph of the Discussion, the authors state, "Back diffusion is prevented because blocks at the exit do not trigger nucleotide exchange." This (rather obscure) statement appears to contradict a later suggestion in the fourth paragraph of the Discussion that a second (?) clamping mechanism by SecA could prevent backsliding.

We see no particular evidence of a clamp in SecA from the MD, and the model does not require one. However, one has been reported previously. We therefore hoped to keep this option open – potentially as a future refinement of the model as more data are obtained in our own labs and by others.

We have clarified this point in the Discussion.

3) In the second paragraph of the Discussion the authors suggest that binding of the signal sequence to SecYEG (or potentially partitioning of the signal sequence into the cytoplasmic membrane) ensures directionality. This insurance would require that the signal sequence be cleaved after the completion of translocation. However, multiple lines of evidence (e.g. Randall and colleagues. Cell 25:151) suggest that the signal sequence is cleaved off during the process of translocation. Wouldn't this make translocation reversible in their model?

The paper in question shows that the cleavage of the signal sequence occurs late in the translocation process. This should make reverse transport unlikely, particularly given the presence of other downstream factors (e.g. chaperone binding or folding of the translocated chain).

We now mention this in the Discussion.

Reviewer #3:

Regarding the molecular dynamics simulations of this study, I think an important opportunity would be missed if the investigators do not look into the simulation data deeper and try to understand--in atomic details the simulations afford – the differential local effects ADP- and ATP-binding give rise to and the allosteric mechanism propagating the effects to the pore. A basic biophysical understanding and the conformational changes in the nucleotide binding sites, presumably involving key conserved residues, should be discussed. (Potentially relevant, there are two analogous papers that show two distinct states ensues ADP-/ATP-binding of a protein kinase [Shan et al., PNAS, 2009; Foda et al., Nature Comm, 2015]).

This is a very good suggestion. In fact, we had already identified a putative conformational switch involved in coupling the nucleotide binding site to the protein channel. We have added a new figure (Figure 3) that highlights the key configurational changes occurring in the immediate vicinity of the nucleotide. Their implications have been discussed in relation to both SecA and the broader helicase family.

Separately, I am concerned of the fastness of the pore closing/opening following placing ADP/ATP in the nucleotide binding site (~20 nanosecond). This is unusually, almost suspiciously fast, considering that typically such conformational changes occur at micro- to millisecond timescale and, because in SEC the information originating for the binding site has to travel nanometers to the pore, I would expect this to take significantly more than several microseconds. It is possible that the crystal structure by serendipity is positioned at a high-energy (possibly even the transition state, which would be very interesting and informative). If so, no barrier-crossing is required, which would be consistent with the nanosecond timescale observed. This should be further examined. Would we qualitatively see the same effect if a different force field (Amber or Charmm) is used? What happens if the nucleotide binding site is unoccupied in the simulations? (The new simulations do not need to run to 1 microsecond each if the conformational change occurs in 10s of nanoseconds).

The points raised by the referee have been addressed above.